# Transplantable human thyroid organoids generated from embryonic stem cells to rescue hypothyroidism

Mírian Romitti [1], Adrien Tourneur [1], Barbara de Faria da Fonseca [1], Gilles Doumont[2], Pierre Gillotay[1], Xiao-Hui Liao[3], Sema Elif Eski[1], Gaetan Van Simaeys[2,4], Laura Chomette[1], Helene Lasolle[1], Olivier Monestier[1], Dominika Figini Kasprzyk[1], Vincent Detours[1], Sumeet Pal Singh[1], Serge Goldman [2,4], Samuel Refetoff[3,5] & Sabine Costagliola [1] ✉

The thyroid gland captures iodide in order to synthesize hormones that act on almost all tissues and are essential for normal growth and metabolism. Low plasma levels of thyroid hormones lead to hypothyroidism, which is one of the most common disorder in humans and is not always satisfactorily treated by lifelong hormone replacement. Therefore, in addition to the lack of in vitro tractable models to study human thyroid development, differentiation and maturation, functional human thyroid organoids could pave the way to explore new therapeutic approaches. Here we report the generation of transplantable thyroid organoids derived from human embryonic stem cells capable of restoring plasma thyroid hormone in athyreotic mice as a proof of concept for future therapeutic development.

Hypothyroidism is a very common disorder with a prevalence of 1–5% worldwide. It results from insufficient thyroid hormone (TH) production due to autoimmune damage to the thyroid gland, iodide excess or deficiency, external irradiation, genetic defects, or other defects present at birth (congenital hypothyroidism, CH) and surgical or radioactive thyroid ablation to treat hyperthyroidism or thyroid cancer[1–3]. Despite well-established TH replacement therapy, it is estimated that up to one-third of patients do not receive an adequate treatment[4,5] while a large proportion have impaired quality of life, particularly psychological well-being[6–8]. In addition, studies have shown that children with CH can develop motor, cognitive, and social dysfunction even when diagnosed through newborn screening followed by early TH replacement[7]. Indeed, constant exogenous supply of TH does not always adapt to changes in TH requirement associated with growth, puberty, pregnancy, and stress, not to mention inconsistent compliance, common in teenagers[9,10]. Regenerative medicine based on iPSC-derived thyroid tissue, might provide a more effective

therapeutic avenue in the future, even if iPSC-derived organ transplantation in humans is currently a long way from clinical applications.

In recent years, significant progress has been made in the development and application of human cell-based models for the study of human biology and for disease modeling. Human embryonic stem cell (ESC)-based protocols enabled the generation of several types of human organoids, including brain, intestine, stomach, liver, kidney, lung, endometrium, prostate, pancreas, and retina[11,12]. Regarding the thyroid, murine ESC-derived organoids have been shown to recapitulate in vitro the developmental stages of the thyroid gland with the ability to produce TH in vitro and in vivo after transplantation into mice with ablated thyroid glands[13–16].

In contrast, human thyroid cells so far generated from stem cells have shown neither full maturation in vitro, nor the ability to compensate for low TH levels when transplanted into animals devoid of thyroid tissue[17–22]. These difficulties in producing functional human thyroid follicles capable of restoring thyroid function in vivo have been

[1]Institut de Recherche Interdisciplinaire en Biologie Humaine et Moléculaire (IRIBHM), Université Libre de Bruxelles (ULB), Brussels, Belgium. [2]Center for Microscopy and Molecular Imaging (CMMI), Université libre de Bruxelles (ULB), Charleroi (Gosselies), Belgium. [3]Departments of Medicine, The University of Chicago, Chicago, IL, USA. [4]Service de Médecine Nucléaire, Hôpital Érasme, Université libre de Bruxelles (ULB), Brussels, Belgium. [5]Departments of Medicine, Pediatrics and Committee on Genetics, The University of Chicago, Chicago, IL, USA. ✉e-mail: sabine.costagliola@ulb.be

partially overcome by using organoids generated from suspensions of fetal and adult human thyroid cells, but with some limitations. Ogundipe et al.[23] generated thyrospheres from human thyroid glands, but 26 weeks are required to detect human thyroid tissue when these organoids are transplanted into hypothyroid mice, and plasma levels of T4 did not increase significantly[23]. Liang et al.[24] demonstrated the derivation of thyroid organoids from human fetal thyroid tissue which secrete T4 in vitro and locally after transplantation[24]. Recently, van der Vaart et al.[25] showed that the treatment of adult tissue-derived organoids with Graves' disease patients' sera lead to increased proliferation and hormone secretion after thyrotropin receptor (TSHR)-antibody stimulation, showing the applicability of the system to model thyroid diseases[25]. These tissue-derived models constitute a great tool to study adult tissue physiology and disease, however, are limited to investigate mechanisms related to early thyroid development/maturation and to provide an alternative to imperfect hormone substitution treatment to restore thyroid function in patients. Thus, there is an urgent need to define an optimized strategy to generate TH-producing human follicles from stem cells. We used forward programming by transient overexpression of *NKX2-1* and *PAX8* transcription factors (TFs) and manipulation of signaling pathways to generate a functional human thyroid from pluripotent stem cells that recapitulates thyroid function in vitro and in vivo. The resulting organoids are characterized with a transcriptomic time course and their functionality is demonstrated in vivo by follicle transplantation in hypothyroid mice.

## Results

### In vitro differentiation of hESCs to human thyroid cells

**hESC line generation and characterization.** In recent years, in vitro mouse ESC-derived thyroid models have shed light on the mechanisms involved in thyroid development and maturation[13–17]. However, replication of these protocols using hESC/iPSC was insufficient to generate a functional human thyroid in vitro[17–21]. Since we have previously shown that forward programming[26–29] by transient overexpression of the transcription factors, *Nkx2-1* and *Pax8*[30] leads to high efficiency of thyroid differentiation and functional follicle formation from mouse ESC[14], we used a similar approach to generate a recombinant human ESC line. First, we took advantage of a previously generated NKX2-1^WT/GFP knock-in hESC line[31] (Supplementary Fig. 1a) to track thyroid differentiation and cell organization using the NKX2-1^GFP reporter. Furthermore, the hESC-NKX2-1^WT/GFP line was modified to allow transient expression of *NKX2-1* and *PAX8*, by adding doxycycline (Dox; 1 µg/ml; Supplementary Fig. 1b) to the culture medium (Fig. 1a). The resulting hESCs, had normal karyotypes and were able to spontaneously differentiate into cells from the three germ layers (Supplementary Fig. 1c, d, respectively).

**Induction of thyroid status.** The modified hESCs were first grown for 2 days in hanging drops to allow the formation of embryoid bodies (EBs) (Fig. 1a). The generated EBs were then cultured in Matrigel (MTG) drops and endoderm was induced by adding Activin A (AA) for 3 days. This treatment increased the mRNA levels of endoderm markers *SOX17* and *FOXA2* compared to the un-induced control (−AA; Supplementary Fig. 1e). It simultaneously improved the ratio of FOXA2⁺ cells, particularly in the inner compartment of the EBs (Supplementary Fig. 1f). After endoderm induction, cells were treated for 4 days with doxycycline (Dox). At day 9, expression of NKX2-1 and PAX8 was detectable by immunofluorescence (Fig. 1b). Flow cytometry quantification demonstrated that approximately 70% of NKX2-1^GFP⁺ cells co-expressed PAX8 while less than 1% of the cells were double positive in the absence of Dox (−Dox condition, Fig. 1c). Furthermore, qRT-PCR analysis showed that not only exogenous *NKX2-1* and *PAX8* gene expression levels were significantly upregulated compared to the −Dox condition, but endogenous *PAX8*, *FOXE1*, *TG*, and *TSHR* mRNA levels were also increased as early as day 9 (Fig. 1d). Since in the present cell line GFP sequence

replaces exon 2 (Supplementary Fig. 1a) of the *NKX2-1* gene, endogenous *NKX2-1* mRNA levels could not be evaluated due to the impaired accuracy of qRT-PCR primers. To determine whether forced overexpression of thyroid TFs leads to autonomous activation of endogenous cell programming towards thyroid fate, Dox treatment was stopped, and cells were incubated in basal differentiation medium for 7 days (from day 9 to day 16). Gene expression analysis of thyroid TFs revealed that exogenous expression of *NKX2-1* and *PAX8* decreased overtime and reached control levels (+AA −Dox) from day 12. On the other hand, endogenous *PAX8* and *FOXE1* levels increased overtime and reached a plateau from day 14. *HHEX* levels were variable and similar to the control condition. This is expected since *HHEX* is known to be highly expressed in the endoderm (Supplementary Fig. 1g). These results suggest that induction of TFs by Dox activates the endogenous transcriptional machinery that initiates thyroid differentiation. In addition, BrdU assays performed from day 9 to day 16 showed a time-dependent decrease in the proportion of proliferating NKX2-1^GFP⁺ cells, indicating an inverse coordination between differentiation and proliferation in the step of thyroid fate commitment of our model (Fig. 1e).

**Thyroid cell population expansion and early differentiation.** Thyroid cell population expansion and early differentiation were promoted by incubation with 8-br-cAMP for 2 weeks (from day 16 to day 30). Flow cytometry analysis confirmed a growth of the thyroid population with approximately 25% of total cells expressing NKX2-1^GFP at day 30 (Fig. 1f and Supplementary Fig. 1). This effect is the product of proliferation promoted by cAMP treatment as evidenced by the increase in NKX2-1^GFP/BrdU⁺ overtime (Fig. 1e and Supplementary Fig. 1i). In addition, transcriptomics analysis performed on NKX2-1^GFP⁺ cells showed an increase in gene expression of early thyroid markers overtime (+AA +Dox +cAMP condition), confirming the role of cAMP in early thyroid cell differentiation (Fig. 1g). This was accompanied by a steady expression of key genes such as *NKX2-1*, *TG*, and *TSHR* from day 23 (Supplementary Fig. 1j). However, key maturation markers, such as *NIS*, *TPO* and *DUOX* family, were not significantly induced by cAMP, suggesting that it was not sufficient to promote thyroid maturation and function (Fig. 1g). By tracking NKX2-1^GFP⁺ cells we observed that at day 28 thyroid cells start to form follicle-like structures and immunostaining shows marked expression of *TG* and *PAX8*. Though, the cells were not organized in single-layered follicles, the presence of a luminal compartment (Supplementary Fig. 1k), suggested that the process of folliculogenesis was initiated but not complete at this stage.

**Promotion of thyroid maturation and function.** Since cAMP treatment did not fully promote thyroid maturation despite significant expression of *TSHR*, we explored additional ways to promote full thyroid differentiation. As the TSHR controls more than the Gs regulatory cascade[32], we first replaced cAMP with hrTSH from day 30. Second, we added dexamethasone (from day 30) and the TGFβ inhibitor SB431542 (from day 37), based on transcriptomics data showing substantial levels of inflammation and TGFβ pathway markers (Supplementary Fig. 2a, b, respectively) among NKX2-1 cells, and the known inhibitory effect of inflammation and TGFβ pathway on thyroid differentiation[33–37]. The switch to hrTSH+Dexa medium showed a marked improvement on *NIS* and *TPO* expression levels when compared to cAMP at day 38 (other controls are also displayed; Supplementary Fig. 2c–g). In addition, the new treatment improved the expression levels of thyroid differentiation markers, while downregulating inflammation markers compared with cAMP-treated cells (Fig. 1g and Supplementary Fig. 2a, respectively). Conversely, the supplementation of the hrTSH+Dexa differentiation medium with SB431542 inhibitor from day 38, resulted in a marked upregulation of *TG*, *NIS* and *TPO* mRNA levels compared to controls at day 45 (Fig. 1g, h and Supplementary Fig. 2h–l, respectively). RNAseq analysis showed that markers of the TGFβ pathway were

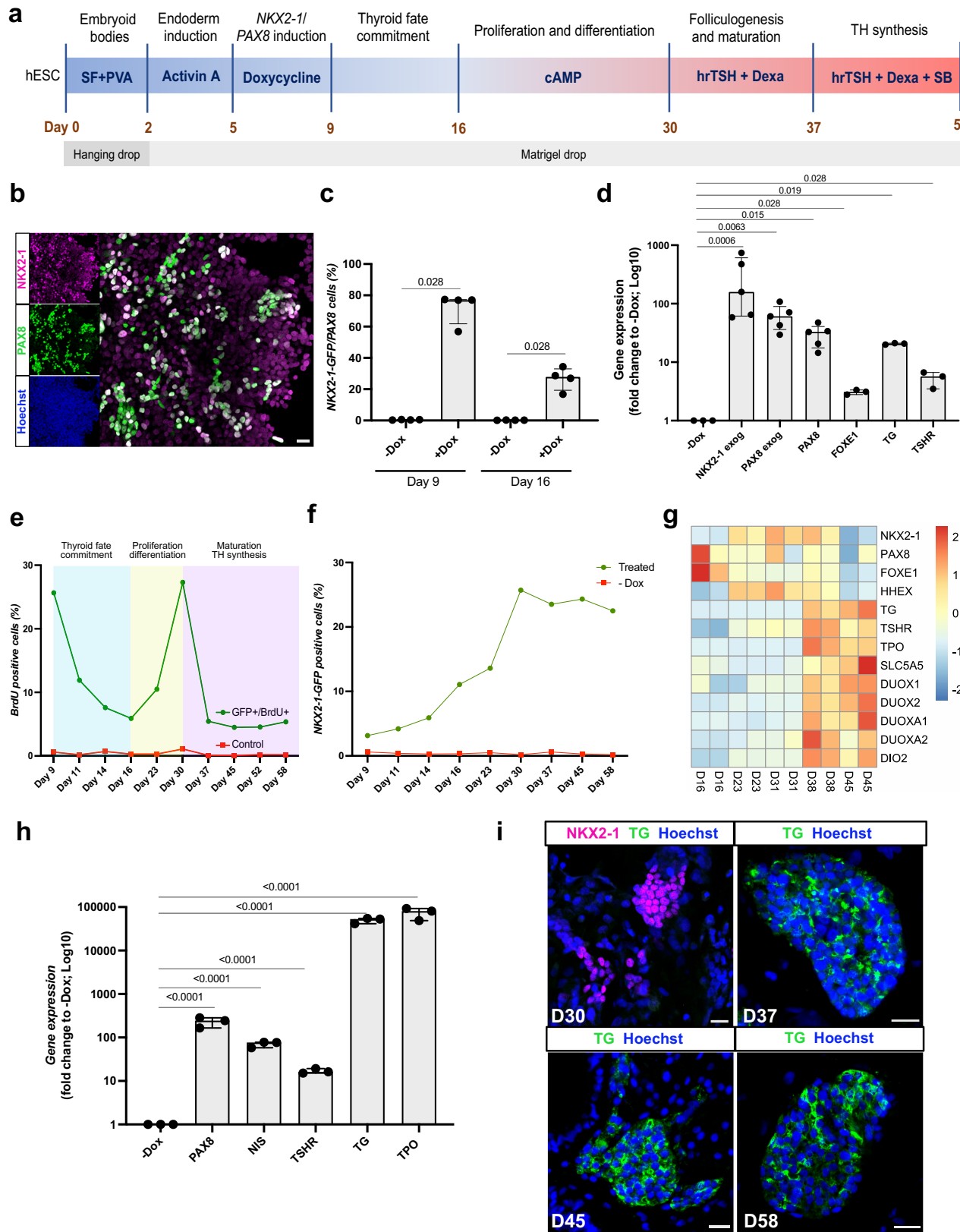

downregulated compared to the cAMP control condition, whereas upregulation of thyroid genes including *DIO2* and the *DUOX* family were also detected (Supplementary Fig. 2b, m and Fig. 1g, respectively). Importantly, although the number of NKX2-1^GFP+/BrdU+ cells decreased significantly after switching to medium supplementation (day 30), the proportion of NKX2-1 cells was preserved overtime, maintaining the cell ratio inside the organoids (Fig. 1f, e, respectively). TG immunostaining was not observed on day 30, but it became detectable from day 37 (Fig. 1i). Additional confocal images from day 45 showed areas of NKX2-1 cells co-expressing E-CADHERIN (Fig. 2a). TG was present in the cytoplasm of these cells, but mainly accumulated in the intra-luminal space (Fig. 2b). TPO

**Fig. 1 | Transient overexpression of *NKX2-1* and *PAX8* promotes differentiation of human ESCs into thyroid follicular cells. a** Schematic representation of the protocol leading to thyroid follicle differentiation from human ESCs. **b** NKX2-1 and PAX8 co-staining at day 9. **c** Quantification by flow cytometry of NKX2-1GFP and PAX8 expressing cells after Dox stimulation, at days 9 and 16 ($n = 4$). **d** qRT-PCR analysis of exogenous *NKX2-1* and *PAX8* and endogenous *PAX8, FOXE1, TG,* and *TSHR* genes after Dox stimulation (day 9) ($n = 5$ for *NKX2-1 exog, PAX8 exog* and *PAX8 endo; n = 3 for FOXE1, TG* and *TSHR*). **e** Proportion of NKX2-1GFP cells expressing BrdU during the differentiation protocol ($n = 3$ per time point). **f** Quantification by flow cytometry of NKX2-1GFP cells during the differentiation protocol ($n = 3$ for −Dox, Day 23, 45 and 58; $n = 4$ for Day 9, 11, 14, 16, 30, and 37). **g** Heatmap of normalized bulk RNA-Seq of thyroid genes expression among NKX2-1GFP cells at different stages of the thyroid differentiation protocol. Rows represent markers and columns represent specific time points. Color values in the heatmap represent mean expression levels. **h** qRT-PCR analysis of *PAX8, NIS, TSHR, TG,* and *TPO* at thyroid organoids from day 45 of differentiation protocol compared to un-induced control (−Dox) ($n = 3$). **i** Immunostaining for NKX2-1 and TG at D30 and TG at day 37, 45, and D58 showing thyroid differentiation and cell organization overtime. Data from at least three independent experiments are shown. Statistical analyses were performed using two-sided unpaired Mann–Whitney (*p* values are presented in the graphs; data presented as median (IQR)). The experiment was performed at least three times with similar results (**b, i**). Scale bars 20 μm. Source data are provided as a Source Data file.

showed cytoplasmatic and apical accumulation (Fig. 2c). Despite the detection of iodinated TG (TG-I) in most follicles, T4 was not detected at this time point (Fig. 2d). Finally, co-staining of PAX8 and ZO-1 further supported the existence of a monolayer-organized structure of follicles with a well-delimited lumen (Supplementary Fig. 2n).

Of note, our human hESC-derived protocol for thyroid generation follows the sequential events observed in vivo. In human, thyroid development takes approximately 40 days from specification to folliculogenesis[38,39]. A similar developmental time is required in our in vitro model. Considering that we artificially induce thyroid progenitor cells formation by transiently forcing the expression of *NKX2-1* and *PAX8*, the expression of maturation genes follows the physiological sequence, with *FOXE1, TSHR* and *TG* being the first detected genes at day 9, followed by increase of their levels and expression of *TPO* and *NIS/SLC5A5*. A similar effect trend was observed in thyroid population expansion, organization and follicle formation. However, even though the TH machinery seems to be complete, we could not detect TH-producing follicles at day 45.

## Single-cell characterization of human thyroid organoids
**Day 45**. To better characterize the resulting cell composition of our hESC-derived thyroid model, scRNA-seq analyzes were performed at day 45. Since the efficiency of the present protocol is approximately 25%, we enriched the proportion of NKX2-1GFP+ cells to 60%, the remaining sorted cells belonging to the GFP- population. A total of approximately 6000 cells were used for scRNA-seq library preparation using the droplet-based assay from 10X Genomics. 1874 cells passed our quality filtering (see methods). These cells were spread among 7 clusters (Fig. 2e, f), including a cluster of thyroid follicular cells with 1176 cells showing expression of genes involved in development and function, including *NKX2-1, PAX8, FOXE1, HHEX, TG, TSHR,* and *TPO* (Fig. 2e–g). Of note, we identified three subclusters among these follicular cells: Thyroid "progenitors" (477 cells) expressing mainly the thyroid TFs; immature thyrocytes (365 cells) also expressing *TG* and *TSHR*; and mature thyrocytes (334 cells) showing a canonical thyroid signature with a higher proportion of *TPO*-expressing cells (Fig. 2e–g and Supplementary Fig. 3a). Furthermore, pseudotime analysis revealed a differentiation trajectory originating from thyroid progenitors, progressing towards immature and, finally, mature thyrocytes (Fig. 2h). Gene expression of thyroid markers along this trajectory followed the expected dynamics, with TFs appearing first, followed by *TSHR, TG,* and *TPO* expression (Fig. 2i).

The identity and molecular signature of cells in the remaining clusters were characterized. We identified four non-thyroidal clusters: fibroblasts (146 cells) expressing *DCN, COL1A2,* and *PPRX1*; cardiovascular cells (182 cells) enriched in *ACTA2* and *TNNT2* markers; airway cells (203 cells) expressing *KRT5* and *TP63* and goblet cells (167 cells) expressing *MUC5AC, MUC5B* and *TFF3* (Fig. 2e, f and Supplementary Fig. 4a, c, e, g). Non-thyroidal mesodermal and epithelial-endodermal cells were characterized by immunostaining. It confirmed the presence of these cells in our organoid system (Supplementary Fig. 4j–l). Of

note, even if NKX2-1 also plays a critical role in lung and forebrain development[34], our protocol predominantly generates thyroid cells: more than 75% of *NKX2-1+* cells co-express *PAX8* and/or other thyroid markers.

**Connectome.** We deployed CellPhone-DB to determine the ligand–receptor interaction pairs jointly expressed in thyroid cell clusters and other cell types. Interestingly, significant cell–cell interactions of thyroid cells mostly involved mesodermal cells and were associated with signaling pathways with known roles in thyroid development and physiology (Supplementary Fig. 5a). Previous studies performed in vivo or using stem cell-derived organoids have described the critical role of BMP and FGF signaling pathways in regulating thyroid specification, as well as how WNT signaling influences commitment to thyroid or lung lineages[13,16,17,40,41]. In our multicellular organoid model, the presence of mesoderm-derived cells might influence thyroid development without supplementation of factors, as we observed that fibroblasts and cardiovascular cells are an important source of *BMP2, BMP4,* and *FGF2* ligands, whereas thyroid cells express the cognate receptors (Supplementary Fig. 5a, b). Insulin-like growth factor (IGF-I) is also known for supporting normal thyroid size and function, in part by enhancing TSH sensitivity[42]. In addition to BMP and FGF, mesodermal, airway and endoderm epithelial cells also provide significant amounts of *IGF-1* and *IGF-2*, while progenitors and immature thyrocytes express *IGF1R* and mature thyrocytes mainly express *IGF2R*. On the other hand, we also observed that fibroblasts and cardiovascular cells express significant amounts of *WNT2, WNT5A, TGFb1,* and *TGFb2*, whereas thyroid cells express their respective receptors (Supplementary Fig. 5a, b). As described previously, in our model, inhibition of TGFβ pathway leads to enhanced thyroid maturation, and this effect may also be related to the repression of such signals from mesodermal-like cells. Immunostaining reveals that the thyroid cells (TG+) are in close contact with mesodermal cells (αSMA+) (Supplementary Fig. 5c), lending further support to these potential interactions.

Despite the cell differentiation and follicular organization observed at day 45, single-cell RNA profiling revealed that a substantial proportion of the thyrocyte population was not fully mature, which may explain the lack of TH detection at this stage. To promote functionality in vitro, we kept the thyroid cells in culture, mimicking the timing of in vivo thyroid maturation.

**Day 58**. Human thyroid development begins around day 20 post-fertilization, while complete organogenesis and TH production can be detected at day 70[38,39]. Based on the time required for thyroid full maturation and TH synthesis in vivo, we cultured the organoids for two additional weeks in the same conditioned medium (Fig. 1a). Another scRNAseq transcriptomic analysis was performed at day 58, following the same protocol steps and conditions as for day 45.

At day 58, 2386 cells met quality control criteria (see methods). We identified 7 clusters (Fig. 3a, b), including a cluster of thyroid follicular cells with 788 cells showing expression of genes involved in

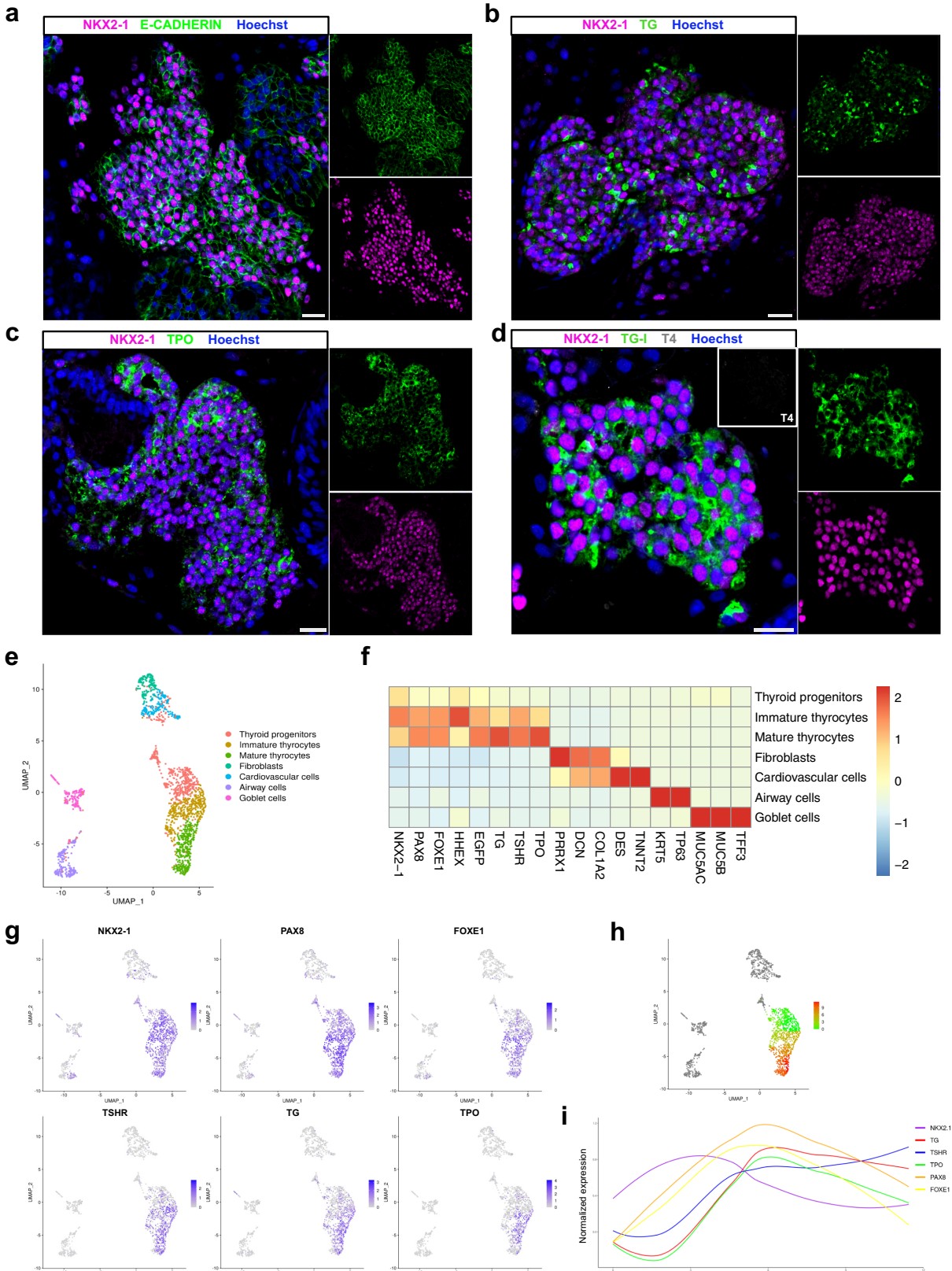

thyroid development and function (Fig. 2a–c). Unlike day 45, here we identified only two thyroid cells subclusters: immature thyrocytes (560 cells) co-expressing the four TFs (*NKX2-1, PAX8, FOXE1, HHEX*), *TG* and *TSHR*; and mature thyrocytes (228 cells) showing a canonical thyroid signature with *TPO*-expressing cells (Fig. 3b, c and Supplementary Fig. 3b). Integration analysis was used to evaluate similarity

between our scRNAseq datasets from day 45 and 58. The analysis revealed a high degree of overlap between immature and mature thyroid cells, and between cardiovascular, airway and goblet cells from the two datasets (Fig. 3d). Furthermore, we compared the similarity of our day 45 and 58 organoids with a dataset of human adult thyroid tissue from the Human Cell Landscape (Fig. 3e)[43].

**Fig. 2 | Immunostaining and Single-cell RNA-seq characterization of human ESC-derived thyroid cells at day 45. a–d** Confocal immunofluorescence images of the three-dimensional follicular structures co-expressing NKX2-1 and (**a**) E-CAD-HERIN, (**b**) TG, (**c**) TPO (**d**) TG-I storage in the lumenal compartment. **e** Single-cell RNA-Seq unsupervised clustering of in vitro hESC-derived human thyroid organoid model cells. Each cluster is represented by a specific color. **f** Heatmap showing normalized expression of selected marker genes with rows representing cell clusters, while columns represent genes. The intensity of the color in each square indicates the mean expression within the cluster. **g** UMAP overlaid with gene expression plots for thyrocyte markers. Color indicates normalized expression. **h** Diffusion analysis of thyrocyte lineage with thyroid progenitor cells as root cells. UMAP overlaid with pseudotime. Color in pseudotime plot indicates order of cell progression. **i** Expression trends of thyroid genes along the pseudotime trajectory. The experiment was performed at least three times with similar results (**a–d**). **a–d** Scale bars, 20 μm.

Interestingly, cells of thyroid organoids and human thyroid tissue clearly overlap (Fig. 3f).

We next characterized the identity and molecular signature of non-target cells. We found five non-thyroidal clusters, 3 of which were already identified at day 45: cardiovascular cells (149 cells) enriched in *DES* and *TNNT2* markers; airway cells (430 cells) expressing *KRT5* and *TP63* and goblet cells (293) expressing *MUC5AC*, *MUC5B* and *TFF3* (Fig. 3b and Supplementary Fig. 4b, d, f, h). Two new clusters that did not show specific signatures of any known cell type were identified. Based on the expression of epithelial markers, such as KRT14 and KRT7, they were labeled "epithelial cells 1" and "epithelial cells 2". The later also expressed AQP5 (Fig. 3b and Supplementary Fig. 4b, i). Immunostaining confirmed the presence of these cell populations in our organoid system (Supplementary Fig. 4m). Of note, the proportion of thyroid cells identified at day 58 was lower than at day 45 as expected given the low proliferation rate of thyroid follicles compared with lung cells (airway and goblet cells population). Also, the lack of a defined cluster of fibroblasts might be due to the long treatment with SB431542, as the TGFβ signaling pathway plays an important role in fibroblast proliferation and maintenance[44,45].

**Connectome.** CellPhoneDB revealed fewer ligand–receptor interactions between follicular and other cell clusters at day 58 compared with day 45 (Supplementary Fig. 5d). Significant cell–cell interactions were observed between thyroid and cardiovascular cells associated with several signaling pathways, such as TGFβ, BMP, FGF, WNT and IGF (Supplementary Fig. 5d, e). To a lesser extent, airway and goblet cells appeared to interact with thyroid cells by producing mainly IGF ligands, whereas these cell populations appeared to interact only with immature thyrocytes via BMP signaling. On the other hand, only airway cells showed expression of WNT ligands interacting with thyroid cells expressing the cognate receptors. Remarkably, the analysis revealed no significant interactions involving epithelial cells 1 and 2. As for day 45, immunostaining for TG and Troponin T (a marker for cardiomyocytes) showed that the thyroid population was in close proximity to the cardiovascular cells supporting a possible interaction between these cells (Supplementary Fig. 5f).

## Complete human thyroid maturation in vitro and thyroid hormone synthesis

Consistent with the improvement in maturation detected by scRNAseq at day 58, RNA expression measured by qRT-PCR confirmed the maintenance of thyroid gene expression levels with a marked increase of *TSHR* mRNA compared to day 45 (Fig. 3g). Immunostaining also showed three-dimensional follicles expressing NKX2-1 and E-CADHERIN (Fig. 3h–h'); TG-expressing follicles presented a defined lumen revealed by F-Actin (Phalloidin) staining (Fig. 3i). In addition, marked TPO staining was observed in most follicular structures, with accumulation of the protein at the apical membrane consistent with increased maturation (Fig. 3j–j'). In vitro THs synthesis was confirmed by TG-I and T4 co-localization within the luminal compartment of hESC-derived thyroid follicles (Fig. 3k). We also evaluated whether thyroid organoids could be cultured over a prolonged period and retain their morphological and functional properties. Immunostaining analysis at day 70 showed that thyroid cells continued to be organized into follicles and to express TG, TPO and TG-I within the lumen (Supplementary Fig. 3c).

## Assessment of in vivo functionality of hESC-derived thyroid follicles

To evaluate the in vivo functionality of hESC-derived thyroid follicles, the recovery of thyroid function was measured in NOD-SCID mice whose thyroid gland was ablated with radioactive iodine (RAI) following low-iodine diet to enhance thyroidal RAI uptake. Thyroid ablation was confirmed after 4 weeks by plasma T4 measurement and SPEC-CT imaging with [123]I. Organoids were harvested and filtered at day 45 to remove most isolated cells, then transplanted under the kidney capsule of RAI mice (Fig. 4a). For the in vivo studies we used 6 non-irradiated (Controls), 6 RAI-ablated and non-transplanted (RAI) and 10 RAI-ablated and transplanted mice (Graft). Macroscopic and histological evaluation of the renal region five weeks after transplantation showed successful implantation of the transplanted organoids in the host niche (Fig. 4b). HE staining showed numerous follicles organized in a manner characteristic of thyroid tissue (Fig. 4b and Supplementary Fig. 6a). The presence of blood vessels in close proximity to thyroid follicles is essential for TH release and its transport to target tissues. Indeed, blood vessels and stromal cells could be observed in the vicinity of the thyroid follicles (Supplementary Fig. 6a–c). Immunostaining for the platelet-derived endothelial cell adhesion molecule CD31 revealed a dense network of small blood vessels surrounding the thyroid follicles, demonstrating the formation of classic angiofollicular units (Supplementary Fig. 6b). The absence of staining overlap between CD31 and Human Nuclear Antigen (HNA) provided unequivocal evidence that the vessels originated from host cells (Supplementary Fig. 6b). On the other hand, the stromal cells were derived from the grafted cells since they co-expressed HNA and alpha smooth muscle actin (αSMA; Supplementary Fig. 6c). HE staining showed that the derived follicular epithelium included both active follicles, which appeared cuboidal to low columnar, and inactive ones, in which the cells were squamous (Supplementary Fig. 6d). Further immunohistochemical analysis supported the formation of functional thyroid follicles (NKX2-1+) at the graft site, including cell polarization labeled by E-CADHERIN, TG cytosolic expression and deposition in the luminal compartment, and the appearance of TPO in the cytoplasm and mainly at the apical membrane (Fig. 4c). T4 immunostaining shows that around 58% of the follicles are active with strong signal in the luminal compartment as observed in adult human tissue (Fig. 4d); nevertheless, the lower proportion of active follicles suggests that the functionality is an ongoing process among the transplanted follicles. However, despite the considerable proportion of functional follicles in the graft, when compared to human adult thyroid tissue, hESC-derived follicles are significantly smaller (87.5 (62.3–120) μm and 49.9 (40.7–68.8) μm, respectively; Supplementary Fig. 6e) which might indicate an early stage of fetal thyroid development[46]. However, comparison of gene expression between the adult thyroid tissue, graft and/or organoids at day 58 showed that the grafted tissue had similar thyroid genes expression compared to human thyroid tissue, whereas the thyroid organoids at day 58 had significantly reduced *TSHR* levels (Fig. 4e). On the other hand, *FOXE1* and *NIS* were upregulated in the organoids compared to adult thyroid tissues (Fig. 4e).

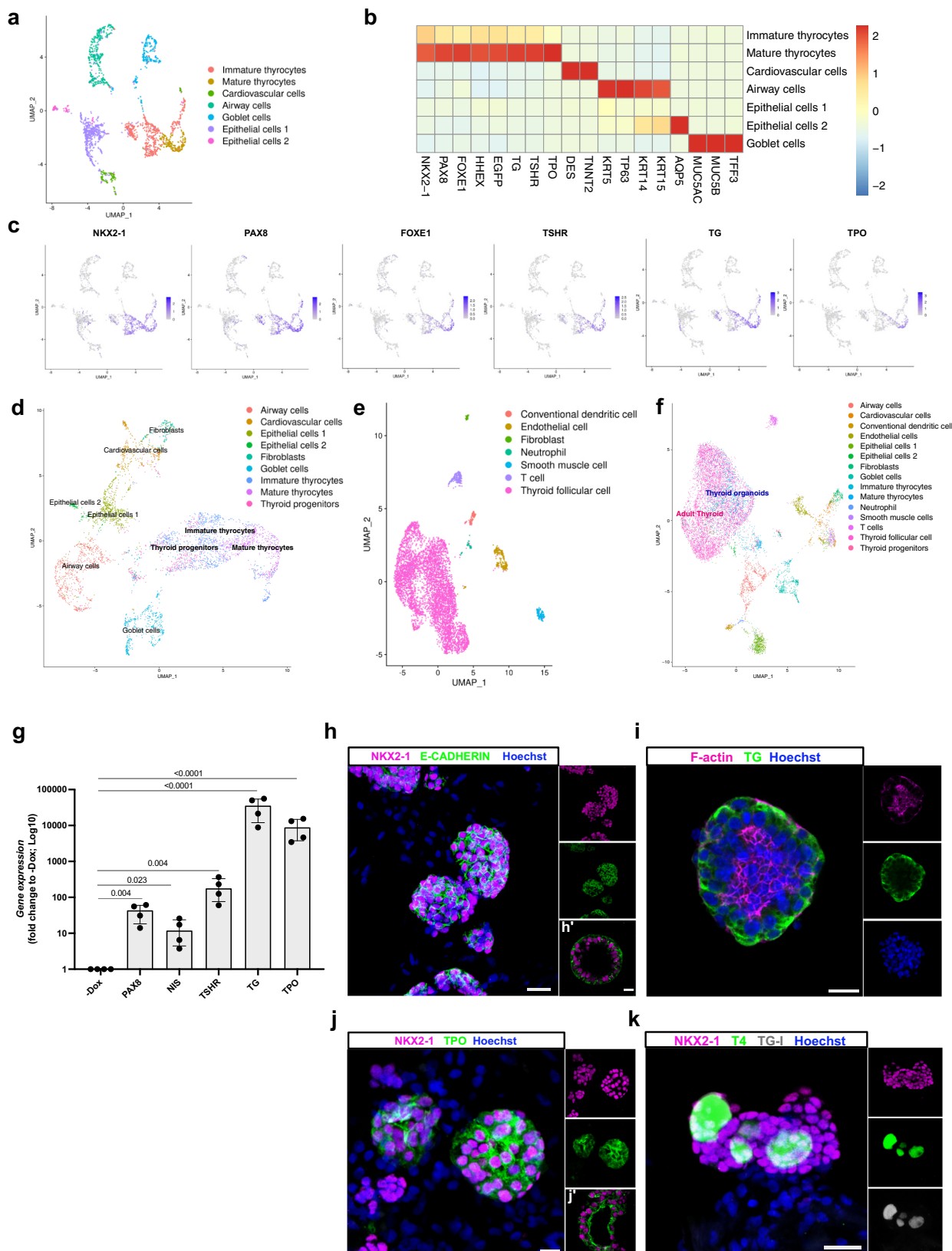

To further evaluate the human thyroid graft performance, we evaluated the ability of NIS-dependent iodide uptake by thyroid tissue with SPECT-CT imaging[47]. Images were acquired four weeks after transplantation and showed a strong uptake signal in the neck (where the thyroid gland is located) of non-ablated mice (Control). In RAI-ablated mice in which organoid were transplanted, no [123]I

uptake was observed in the neck, but a very strong signal was detectable at the site of transplantation, near the kidney (Fig. 4f). After 5 weeks, we evaluated the thyroid function and observed a marked increase in plasma T4 levels among transplanted animals (graft) compared with barely detectable levels in RAI-ablated and non-transplanted mice. These levels, however, were lower than in

**Fig. 3 | Single-cell RNA-seq and immunostaining characterization of human ESC-derived thyroid cells at day 58. a** Single-cell RNA-Seq unsupervised clustering of in vitro hESC-derived human thyroid organoid model cells. Each cluster is represented by a specific color. **b** Heatmap showing normalized expression of selected marker genes with rows representing cell clusters, while columns represent genes. The intensity of the color in each square indicates the mean expression within the cluster. **c** UMAP overlaid with gene expression plots for thyrocyte markers. Color indicates normalized expression. **d** UMAP showing integration analysis of single-cell RNA-seq data from thyroid organoids at day 45 and 58. **e** Single-cell RNA-Seq unsupervised clustering of publicly available human adult thyroid tissue dataset. Each cluster is represented by a specific color. **f** UMAP showing integration analysis of single-cell RNA-seq data from human adult thyroid tissue and hESC-derived

thyroid organoids at days 45 and 58. **g** qRT-PCR analysis of *PAX8, NIS, TSHR, TG*, and *TPO* at thyroid organoids from day 58 of differentiation protocol compared to un-induced control (−Dox) (*n* = 4). **h**–**k** Confocal immunofluorescence images at day 58 of the differentiation protocol. **h** Three-dimensional follicular structures co-expressing NKX2-1 and E-CADHERIN. **i** Intra-lumenal F-actin (Phalloidin) and TG; (j) NKX2-1 cells with TPO cytoplasmic and apical membrane accumulation. **k** NKX2-1 cells showing TG-I and T4 stored in the lumenal compartment. Data from at least three independent experiments are shown. Statistical analyses were performed using two-sided unpaired Mann–Whitney (*p* values are presented in the graphs; data presented as median (IQR)). The experiment was performed at least three times with similar results (**h**–**k**). Scale bars, 20 μm and 10 μm for high magnification follicles. Source data are provided as a Source Data file.

the control group (2.00 (0.71–2.77), 0.27 (0.20–0.35) and 4.72 (4.30–5.52) μg/dl; respectively; Fig. 4g). Yet, T4 levels increased during the week 4 to week 5 interval (Supplementary Fig. 6f). This progression suggests that a complete rescue could occur at later time points. Interestingly, we observed that T3 levels among grafted mice were induced at levels similar to those of the control group, while they were significantly reduced in RAI-ablated mice (36.40 (31.08–73.68), 42.45 (37.85–52.35) and 3.50 (3.00–5.55) μg/dl; respectively; Fig. 4h).

Finally, we evaluated the systemic impact of the organoid grafts. Along with the restoration of THs plasma levels, grafted mice also showed a decrease in plasma TSH levels, which was inversely correlated to T4 (*r* = −0.86; Fig. 4i). Studies have shown that modulation of thyroid status leads to changes in type 1 deiodinase (*Dio1*; D1) levels in the liver[48–50]. To assess systemic recovery, *Dio1* mRNA levels were measured in the liver of the grafted, RAI-ablated and control mice. Notably, hypothyroid mice (RAI) had significantly lower levels than controls (0.0069 (0.0056–0.0090) A.U. and 3.471 (1.58–6.14) A.U., respectively), which was partially offset by transplantation of hESC-derived thyroid follicles (0.177 (0.032–0.709) A.U.; Fig. 4j). In addition, *Dio1* mRNA levels strongly correlated with T4 and T3 levels (*r* = 0.84 and *r* = 0.71; Supplementary Fig. 6g, h, respectively).

## Discussion

The generation of functional human thyroid follicles from hESCs in vitro has proven to be extremely challenging, compared to mouse-derived equivalents. Although the generation of thyrocytes or thyroid follicles from human stem cells has been reported[17–21], these human thyroid cells or thyroid follicles have neither shown full maturation in vitro, nor the ability to compensate for low TH levels when transplanted into animals devoid of thyroid tissue. Here we present a novel protocol for the generation of hESC-derived thyroid organoids able to synthetize TH in vitro and in vivo.

Challenges to translate mESC-derived models to human might be due to the critical differences in pluripotency and gene expression dynamics between mouse and human ESCs. mESCs can be categorized as naïve (derived from inner cell mass (ICM)) and primed or mEpiSCs (derived from epiblast stem cells)[51–55]. They mostly differ in their developmental potential: only naïve mESCs can efficiently contribute to blastocyst chimeras, whereas primed SCs have the potential to differentiate into primordial germ cells in vitro[54,56–59]. Human ESCs resemble primed mEpiSCs with respect to pluripotency, morphology, genetic, transcriptomic and epigenetic status, suggesting a later developmental stage in vivo[52–54,60–62]. Comparative studies have shown differences in the initial expression dynamics of transcription factor and stimulus response genes, which might explain why some differentiation processes unfold differently from mESC and hESC[61]. Recent studies indicate that genetic modifications and optimization of culture systems fosters induction of a more naïve state in hESC[56,57,63–65] which is critical to improve applications and efficiency/success in generating models in vitro.

The generation of organoids with high efficiency is essential to explore mechanisms related to developmental biology, disease progression and therapies. Thyroid organoids have been generated from the fetal and adult thyroid tissue of rats and humans[23–25,66,67]. Those protocols result in three-dimensionally organized pure population of follicular structures with high degree of differentiation, iodine uptake capability which results in functional activity. Due to the high similarity to the original tissue, these organoids provide a robust system to study mechanisms related to adult thyroid diseases, as demonstrated for thyroid cancer (mouse) and Graves' disease modeling (human)[25,66]. An important limitation of organoids derived from thyroid tissue is that the procurement of normal human tissue limits therapeutic applications.

Directed differentiation and transient expression of transcription factors are common methods to promote differentiation of pluripotent stem cells into various mature cell types[14,16,26–29]. Since mature thyroid cells are known to be poorly proliferative[68] a large pool of progenitor cells is required to obtain sufficient thyrocytes for developmental studies and transplantations experiments. Mouse thyroid organoids have been generated by both methods and progenitor cells derivation efficiency was significantly higher when *Nkx2-1* and/or *Pax8* were transiently induced (60–80%)[14,16] compared to exclusive manipulation of signaling pathways such as BMP4 and FGF2 pathways (18%)[17,20]. In addition, to accommodate for their lower efficiency, protocols for directed thyroid differentiation include a step of cell sorting to enrich the thyroid cell population before cell maturation. In the present protocol, the multicellular organoids were kept in culture until the end of the protocol to achieve efficient in thyroid cells generation and avoid purification, as we have previously demonstrated with mouse ES cells[14,37]. This is strong evidence that non-thyroidal cells provide critical signals for full maturation of the thyroid organoids in vitro and for their functionally productive integration in the host in vivo.

Despite the growing knowledge of the molecular mechanisms and signals involved in thyroid organogenesis, little is known about the role of cell–cell interaction at different stages of thyroid formation. For example, studies in zebrafish have shown that the thyroid gland is in intimate contact with the distal ventricular myocardium at early stages of development[69]. In addition, cardiac tissue appears to be an important source of BMP and FGF signals, which are known to promote thyroid specification[13,16,17,40]. Similarly, signaling through IGF-1R is essential for normal thyroid formation and function, particularly by supporting thyroid cell survival through TSHR signaling[42,70]. IGF-1R and insulin receptor (IR) double-thyroid knockout mice showed hypoplastic thyroid glands at birth, folliculogenesis defects and specifically repressed *Foxe1* expression[71]. Besides, in vitro studies suggest that IGF-1 is required for the differentiation of mouse embryonic stem cells into thyroid epithelial cells, supporting the role of this signal in thyroid development[16,17]. Interestingly, our hESC-derived multicellular organoids also contain mesodermal (cardiomyocytes and fibroblasts), airway and epithelial cells, which show the potential for cell interaction with the thyroid population by producing factors BMP, FGF and IGF-1,

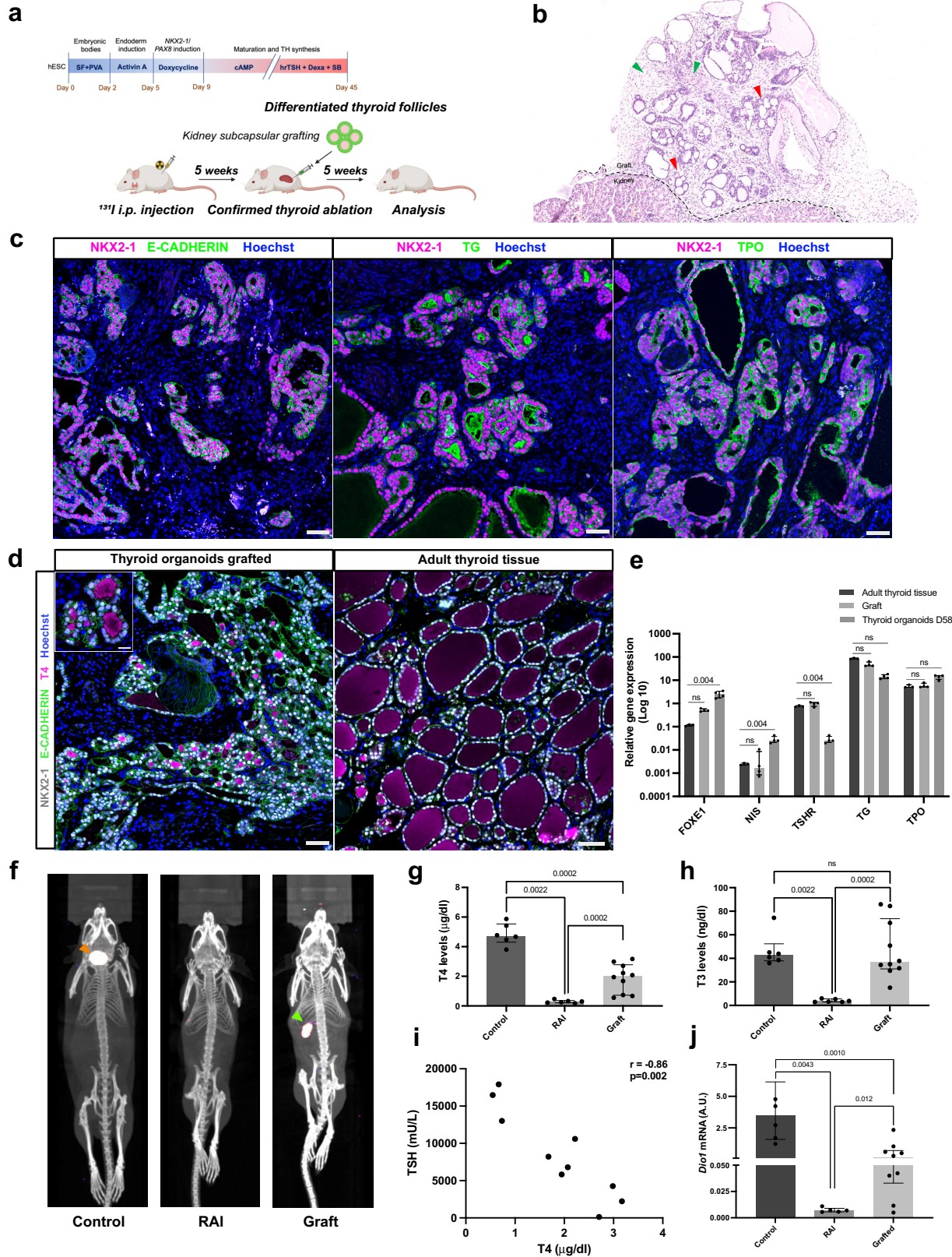

known to play a key role in efficient and proper thyroid development and function in vitro.

In addition to cardiovascular cells, a substantial population of TGFβ-producing fibroblasts was present in our thyroid organoids. TGFβ inhibits thyroid function[34,35,37]. We have shown here that inhibition of the TGFβ pathway contributes to improve thyroid maturation.

On the other hand, fibroblasts are an important source of angiogenic factors such as VEGF and FGF2[72,73] and therefore promote endothelial cells migration and survival[74] within the grafted tissue, allowing the formation of a vascular network supporting graft survival and function. These fibroblasts are also major contributors to extracellular matrix deposition[75] which is instrumental to the maintenance of

**Fig. 4 | In vivo functionality of transplanted human ESC-derived follicles. a** Schematic representation of the hESC-derived thyroid organoids transplantation protocol under the kidney capsule of NOD-SCID thyroid RAI-ablated mice. **b**, **c** Histological analysis of the grafted tissue 5 weeks after transplantation. **b** Hematoxylin and eosin staining shows human thyroid follicles located in the cortical region of the host kidney. Red arrows show the monolayer epithelium of the transplanted tissue surrounded by stromal cells (green arrows). **c** Confocal images show co-expression of NKX2-1 and E-CADHERIN in the monolayered follicles. The grafted tissue shows TG mainly accumulated in the luminal compartment, whereas TPO is strongly expressed in the apical membrane. **d** NKX2-1, E-CADHERIN and T4 immunostaining demonstrate hESC-derived follicles with specific accumulation of T4 within the lumen of several structures as observed in human adult thyroid tissue. **e** qRT-PCR analysis of *FOXE1, NIS, TSHR, TG* and *TPO* expression from human adult thyroid tissue sample compared to graft tissue and/or hESC-derived thyroid cells from day 58 (normalized by *PAX8* levels; n = 3, 4 and 4, respectively).

**f** Maximum intensity projections generated from SPEC/CT images of controls (left), RAI-ablated and non-transplanted (middle), and RAI-ablated and transplanted (right) mice. Images were obtained four weeks after organoids transplantation. The $^{123}$I uptake in the mouse thyroid tissue is shown by the orange arrow, while the signal from the human thyroid tissue (graft) is highlighted by the green arrow. **g**, **h** Plasma levels of (**g**) T4 and (**h**) T3 in controls (n = 6), RAI-ablated (RAI; n = 6) and RAI-ablated/grafted mice (graft; n = 10). **i** Correlation of plasma TSH and T4 levels among grafted animals (n = 10). **j** Comparison of hepatic *Dio1* mRNA levels in controls (n = 6), RAI-ablated (n = 6) and RAI-ablated/transplanted mice (n = 10). All measurements were performed five weeks after transplantation. Two-sided unpaired Mann–Whitney (**g**, **h**, **j**) and Spearman Correlation (**i**) tests were used for statistical analysis (r and p values are presented in the graphs; ns not significant; data are presented as median (IQR)). Similar results were obtained from all grafted samples (**c**, **d**). Scale bars, 50 μm and 20 μm for zoomed area. Source data are provided as a Source Data file.

cellular contacts between follicles and the formation of the angiofollicular units required for thyroid hormones secretion into the bloodstream.

Recent studies have shown that human thyroid organoids generated from thyroid tissue (pure thyroid population)[23,24] have limited efficiency after transplantation. Liang et al. demonstrated only local T4 after transplantation[24]. Furthermore, Ogundipe et al. showed that it takes at least 26 weeks for human thyroid tissue to be detectable when transplanted into hypothyroid mice, while plasma T4 levels were not significantly increased[23]. In addition, the study describes a limited presence of blood vessels surrounding the thyroid follicles. In contrast, we observed a dense network of small blood vessels and the subsequent formation of full-fledged angiofollicular units 5 weeks after transplantation[23]. It is therefore very likely that the presence of stromal cells in the population of grafted cells contributed greatly to the success of our transplantation experiments.

Thus, the present study is the first to show that hESC-derived thyroid follicles produce TH in vitro and in vivo after transplantation into thyroidectomized mice. This model opens a new window on the morphogenetic processes and gene regulatory networks associated with thyroid development. In addition, it could be used as a model to study the role of newly identified variants in causing congenital hypothyroidism and to test the toxic effects of many compounds to which we are constantly exposed in our daily life, in particular endocrine disruptors[76], an emerging challenge in thyroid pathophysiology. Although this model still needs improvement before it becomes therapeutically applicable, it provides a proof of concept that generating autologous human thyroid tissue to maintain TH levels is within reach.

## Methods

### Ethics oversight
Our research complies with all relevant ethical regulations. Human studies were approved by the Ethics Committee of Erasme Hospital, Université Libre de Bruxelles (ULB). Histologically normal human thyroid tissue was obtained from a patient undergoing thyroidectomy. The recruitment was based on patient agreement and availability of the tissue (Hopital Erasme-ULB ethics committee P2016/260). Animal studies were approved by the Commission d'Ethique du Bien-Être Animal (CEBEA) Faculté de Médecine ULB, Project CMMI-2020-01.

### Generation of tetracycline-induced hESC line
The human embryonic stem cell line HES3 used in this study was genetically modified at the NKX2-1 locus to allow insertion of sequences encoding green fluorescent protein (GFP), resulting in the NKX2-1$^{wt/GFP}$ hESC line as previously described[31] (Supplementary Fig. 1a). To generate an inducible NKX2-1-PAX8 hESC line, we cloned the coding sequences of the *NKX2-1* and *PAX8* genes, separated by an IRES sequence, into the pInducer20 lentiviral vector (a gift from

Stephen Elledge; Addgene plasmid # 44012; RRID: Addgene_44012), which contains the sequences for the TRE/rtTA-inducible system (Supplementary Fig. 1b). Lentiviral supernatants were generated by transient transfection of HEK293 cells according to Lipofectamine ™ 2000 (Invitrogen) transfection protocols and harvested 48 h after transfection. To promote the integration of sequences into the genome of the NKX2-1$^{WT/GFP}$ HES3 line, hESCs were plated at high density (1:3) in a Matrigel-coated 6-well culture dish and infected with 50 μl of lentivirus supernatant and 6 μg/ml polybrene for 18–20 h in mTeSR medium (Stem Cell). Positive clones were selected with 300 μg/ml neomycin (Invitrogen). Clones were treated with 1 μg/ml doxycycline (Sigma) for 48 h and screened by immunostaining against NKX2-1 and PAX8 to verify transgene expression. Selected clones were tested for genomic integrity using the G-banding technique according to the protocol described previously[77]. Pluripotency was assessed by testing the ability of the clones to differentiate into cells from the three germ layers. Cells were cultured in basal differentiation medium (Supplementary Table 1) for 21 days and the formation of endoderm, mesoderm and ectoderm cells was assessed by immunofluorescence staining against AFP, αSMA and β-III tubulin, respectively. The hESC-NKX2-1-PAX8 line was registered and approved by the European Human Pluripotent Stem Cell Registry (hPSCreg) as ESIBIe003-A-6.

### hESC culture and differentiation
Modified hESCs were cultured and propagated on Matrigel-coated 6-well culture dishes in Stem Flex medium (Thermo Scientific) supplemented with 100 U/ml Penicillin-Streptomycin (Gibco). For the generation of embryoid bodies (EBs), highly confluent hESCs were detached with 0.5 mM EDTA solution and diluted with 100,000 cells/ml in Stem Flex medium supplemented with 4 mg/ml polyvinyl alcohol (PVA; Sigma) and EBs formation was induced as previously described[14,26]. Briefly, hESCs (2000 cells per droplet) were cultured in hanging drops for two days, then EBs were collected and embedded in growth factor-reduced Matrigel (BD Biosciences); 50 μl Matrigel drops (containing approximately 20 embryoid bodies per drop) were replated onto 12-well dishes. Embryoid bodies were differentiated and cultured in differentiation medium containing DMEM/F12 + Glutamax (Gibco) with 20% FBS (Gibco), 0.1 mM non-essential amino acids (Gibco), 1 mM sodium pyruvate (Gibco), 0.1 mM 2-mercaptoethanol (Sigma), 100 U/ml Penicillin-Streptomycin (Gibco), and 50 μg/ml L-ascorbic acid (Sigma). Cells were supplemented with 50 ng/ml Activin A (Cell GS) for three days to induce foregut endoderm. Expression of *NKX2-1* and *PAX8* was induced by incubation with 1 μg/ml doxycycline (Dox; Sigma) for four days. Cells were then cultured in basal differentiation medium for one week to allow expansion of thyroid progenitors, while differentiation and maturation were induced by treatment with 300 μM 8-br-cAMP (Biolog Inc.), 1 mU/ml rhTSH (Genzyme), 50 nM dexamethasone (Dexa; Sigma) and 10 μM SB431542

(Peprotech) where indicated (Fig. 1a). Culture medium was changed every 48 h.

## NKX2-1$^{GFP}$ population assessment and proliferation assay—flow cytometry

hESCs under thyroid differentiation protocol were incubated with 10 μL/ml of 1 mM BrdU in culture medium for three hours, at several time points. Then, treated cells and controls were washed once with PBS, collected and prepared for flow cytometry immunostaining as follows: Matrigel drops (at least 4 samples per time point) were first digested with HBSS solution containing 10 U/ml dispase II (Roche) and 125 U/ml collagenase type IV (Gibco, Thermo Fisher) for 30–60 min at 37 °C; then a single-cell suspension was obtained by dissociation with TripLE Express (Thermo Fisher) for 10–15 min incubation at 37 °C, the enzymes were inactivated by addition of differentiation medium. After centrifugation, samples were rinsed with PBS, fixed and stained following the APC BrdU Flow Kit protocol (BD Biosciences). In addition, PAX8 antibody was used to stain the thyroid cells. NKX2-1$^{GFP}$, PAX8 and BrdU incorporated cells (BrdU + ) data were obtained and processed using an LSR-Fortessa X-20 flow cytometer and FACSDiva software (BD Biosciences). Unstained cells and isotype controls were included in all experiments. In addition, the percentage of NKX2-1$^{GFP}$ cells was used to estimate the thyroid generation efficiency of our protocol. Also, NKX2-1$^{GFP}$/PAX8 cells were used to evaluate the thyroid fate commitment. Gate strategies are exemplified in Supplementary Fig. 1h–i.

## RNA extraction and quantitative real-time PCR

For total RNA extraction, human organoids (at different time points; At day 45 and 58 follicles enriched samples were used for the analysis), in vivo samples, and human thyroid tissue (histologically normal thyroid tissue was obtained from a patient undergoing thyroidectomy; Hopital Erasme-ULB Ethics Committee approval; P2016/260), were lysed using RLT lysis buffer supplemented with 1% 2-mercaptoethanol (Sigma), and RNA isolation was performed using the RNeasy micro kit (Qiagen) according to the manufacturer's instructions. For reverse transcription, the Superscript II kit (Invitrogen) was used, and qPCR was performed in triplicates using Takyon (Eurogentec) and CFX Connect Real-Time System (Biorad). Results are presented as linearized values normalized to housekeeping gene, GAPDH (human) or β2-microglobulin (mouse) and the indicated reference value (2-ΔΔCt). Specifically, to compare mRNA levels among adult thyroid tissue, graft tissue and organoids from day 58, *PAX8* levels were used for normalization, to overcome the cell heterogeneity among the samples. Gene expression profile was obtained from at least three independent experiments. Primer sequences are shown in Supplementary Table 2.

## RNA-seq and analysis of bulk samples

Bulk RNA-seq was performed in hESC-differentiated cells every week from day 16 to day 45 of our differentiation protocol (Fig. 1a). For each sample, at least 6 wells were pooled together to overcome the variability among experimental wells. In addition, samples treated only with cAMP, collected at day 38 and day 45, were also sequenced and used as control to evaluate the effect of replacing cAMP by hrTSH, Dexa and SB431542 on thyroid maturation. The NKX2-$^{GFP+}$ cell population was obtained by FACS sorting (FACS Aria; BD Bioscience) after sample preparation was performed as previously described (section "NKX2-1$^{GFP}$ population expansion assessment and proliferation assay -flow Cytometry"). In brief, 10,000 NKX2-1$^{GFP+}$ cells were directly sorted into 700 μl of Qiazol lysis reagent (Qiagen) and RNA isolation was performed using the miRNeasy micro kit (Qiagen) according to the manufacturer's instructions. RNA concentration and quality were evaluated using Bioanalyser 2100 (Agilent) and RNA 6000 Nano Kit (Agilent). RNA integrity was preserved, and no genomic DNA contamination was detected. Ovarion Solo RNA-seq Systems (NuGen) was used as indicated by the manufacturer, resulting in high-quality

indexed cDNA libraries quantified with the Quant-iT PicoGreen kit (Life Sciences) and Infinite F200 Pro plate reader (Tecan); DNA fragment size distribution was examined with the 2100 Bioanalyzer (Agilent) using the DNA 1000 kit (Agilent). Multiplexed libraries (10ρM) were loaded onto flow cells and sequenced on the HiSeq 1500 system (Illumina) in high-output mode using the HiSeq Cluster Kit v4 (Illumina). Sequenced data were uploaded on the Galaxy web platform version 22.05.1 and the public server https://galaxyweb.eu was used for mapping and counting analysis. Approximately 10 million paired-end reads were obtained per sample. After the removal of low-quality bases and Illumina adapter sequences using Trimmomatic software[78], sequence reads were aligned against the human reference genome (GRCh38) using HiSat2 software[79]. Raw reads were determined with HTSeq software[80] using the Ensembl genome annotation GRCh38.p13. Normalization and differential expression analyze were performed with two biological replicates (each containing at least 3 distinct wells) per sample using the website iDEP version 0.93[81]. Genes for which expression values were lower than 5 were filtered out. The fold changes of mean gene expression for the duplicates were used to calculate the level of differential gene expression.

## Single-cell RNAseq characterization of thyroid organoids

Cells originating from human thyroid differentiation protocol, at day 45 and day 58, were isolated for scRNAseq profiling, following the procedures previously described[37]. Single-cell suspension preparation and FACS cell sorting were performed as previously mentioned ("NKX2-1$^{GFP}$ population expansion assessment and proliferation assay - flow cytometryn" and RNA-seq and analysis of bulk samples sections). Different proportions of viable NKX2-1/GFP+ (60%) and NKX2-1/GFP− (40%) cells were sorted to guarantee the representation of the distinct cell types present in the organoid culture. Sorted cells were collected in PBS at a density of 800 cells/μl and diluted accordingly to kit's instruction (10x Genomics Chromium Single Cell 3' v3). Around 6000 cells were loaded onto a channel of the Chromium Single Cell 3' microfluidic chip and barcoded with a 10X Chromium controller followed by RNA reverse transcription and amplification according to manufacturer's recommendations (10X Genomics). Library preparation was performed based on 10X Genomics guidelines. Libraries were sequenced using Illumina NovaSeq 6000 system.

## Single-cell RNAseq data analysis

Raw sequencing data from both time points (D45 and D58) were aligned, annotated, demultiplexed and filtered using Cell Ranger Software (v.6.0.1) with a custom-built reference. The custom-built reference was based on the human reference genome GRCh38 and gene annotation Ensembl 98 in which the EGFP sequence was included. The new reference was generated using the cellranger mkref function from the Cell Ranger Software. Analyses were done using R 4.1.0 and Seurat version 4.0.3[82]. Briefly, raw counts from Cell Ranger were loaded and the "background soup" was removed using SoupX[83]. The background soup refers to ambient RNA molecules contaminating cell-containing droplets, a common problem in droplet-based single-cell RNA-sequencing technologies. Decontaminated UMIs were then filtered to discard any doublet (droplet containing two cells instead of one) using DoubletFinder[84]. Finally, cells containing less than 200 unique genes or more than 26% of UMI counts related to mitochondrial genes were discarded. The 26% threshold was selected to discard dying cells while retaining as much barcodes as possible. The resulting library was scaled and normalized using the SCTransform function from Seurat. For the day 45 dataset cell cycle effects and mitochondrial content were used as variables to regress out with SCTransform. For the D58 dataset only mitochondrial content was regressed out. Principal component analysis (PCA) was computed using the 3000 first variable features, and the top 30 principal components were used for SNN graph construction, clustering (resolution 1) and UMAP

embedding using Seurat's functions and recommended methods. Cluster annotation was based on marker genes obtained using Seurat's FindAllMarkers function and literature survey. Pseudotime analysis in thyroid populations was performed using Monocle3[85] with default parameters and with data imported from the Seurat object, selecting thyroid progenitors as root cells. Pseudotime-related plots were generated using the FeaturePlot function from Seurat and the geom_smooth function from ggplot2. Receptor-ligand interaction analysis was done with CellPhoneDB, which consists in a public repository of ligands, receptors and their interactions enabling a comprehensive and systematic analysis of cell–cell communication[86]. CellphoneDB was run using the statistical method with default parameters. A manually selected list of biologically relevant ligand–receptor pairs displaying statistically significant interaction was used to create the dot plot showing the interactions of thyroid populations with other cell populations. Human thyrocyte expression counts were downloaded from the human cell landscape website and processed using the Seurat SCTransform pipeline described above with mitochondrial regression only. One of these two datasets composed mainly of immune cells was discarded as its abnormal composition prevented useful comparisons with our organoid data that do not include immune cells. Integration of D45, D58 and the public human cell landscape dataset was done using Seurat's reciprocal PCA integration with 30 dimensions, following Seurat's guidelines.

### Follicles enrichment for in vivo transplantation

Thyroid organoids at day 45 of differentiation were washed twice with Hanks's balanced salt solution (HBSS, containing calcium and magnesium; Gibco), then 1 ml of a digestion medium containing 10 U ml dispase II (Roche) and 125 U ml of collagenase type IV (Sigma) diluted in HBSS was added to each well. The organoids were carefully removed using a 5 ml pipette and transferred to a sterile Erlenmeyer and incubated at 37 °C in a water bath with shaking for 45–60 min. The release of thyroid follicles was tracked by microscopy (bright field and GFP). When isolated structures were detected, enzymes were inactivated by the addition of 10% FBS followed by centrifugation at 500 g for 3 min. Cells were rinsed twice with HBSS and the follicles population was enriched using 30 μm (single-cell removal) and 100 μm (follicles enrichment; 30–100 μm size) reverse strainers (Pluriselect). Finally, the 3D structures were counted and approximately 10,000 structures were resuspended in 65 μl of differentiation medium for in vivo transplantation.

### RAI-induction of hypothyroidism, transplantation of hESC-derived thyroid follicles and SPECT-CT imaging

All animal experiments were performed in accordance with local Animal Ethics (Commission d'Ethique du Bien-Être Animal (CEBEA) Faculté de Médecine ULB, Project CMMI-2020-01). A cohort of 16 five-week-old female non-obese and non-diabetic mice with severe combined immunodeficiency (NOD -SCID) (Charles River Laboratories, France) was placed on an iodine-deficient diet for one week after arrival.

In addition, six NOD -SCID mice, not submitted to any treatment were included in the study as untreated controls. One week after starting the diet (first week), 16 mice were injected intraperitoneally with approximately 5.75 MBq [131]I (GE Healthcare Belux, Belgium) in 90 μL volume supplemented with 10 μL NaCl 0.9% (MiniPlasco, BBraun). To confirm thyroid gland ablation by [131]I, SPECT-CT images with Na[123]I were obtained on a nanoSPECTPlus (for the SPECT) and a nanoScanPETCT (for the CT) (Mediso, Hungary) equipped with a Minerve rat cell implemented with a mouse insert. In the fourth week, mice were injected intravenously with 8.75–9.33 MBq [123]I 24 h before imaging. SPECT/CT imaging was performed on two mice in parallel under isoflurane anesthesia (1.8% isoflurane, 2.0 l/min O2) with the following parameters: collimator aperture APT105, 'fast' helicoidal

acquisition mode with a duration of 50 s/projection to acquire 1000 counts per projection, scan range of 105 mm, reconstruction in standard mode, i.e. 35% smoothing, 3 iterations and 3 subsets to obtain a voxel size of 750 μm$^3$. CT was performed with the following parameters: 480 projections, minimum zoom, binning 1:4, 50 kV, 300 ms/proj, scan range of 115 mm. Acquisition data were reconstructed with a Feldkamp-based algorithm generated to obtain a cubic voxel of 250 μm$^3$, using a cosine filter with a cut-off of 100%. Then, one week later (week five) 10 RAI-ablated mice were transplanted with thyroid organoids. Mice were first treated with 0.01 mg/ml − 50 μl Temgesic (Schering Plow), anesthetized under isoflurane anesthesia, and the eyes/cornea were protected with Vidisic gel (Bausch & Lomb Inc.). Mice were injected with 8 μl of follicle-enriched suspension of thyroid organoids (described in "Follicles enrichment for in vivo transplantation") under the capsule of one kidney using a 30 G needle syringe (Hamilton Bonaduz AG) (the kidney was exposed through skin/muscle/peritoneum incision via a dorsolateral approach). The entire cohort of mice had blood collected 4 weeks after transplantation (week 9) and was imaged as described above to assess the iodine uptake capacity of the transplanted tissue. At the end of the experiment (week 10), control, RAI-ablated and grafted mice were analyzed. Mice were finally sacrificed, blood was collected for T4, T3 and TSH analyses, liver for *Dio1* mRNA measurement, while the kidney containing the transplanted tissue was harvested for transcriptomic and histological analyses. Qualitative analysis of the images was performed using VivoQuant v3.5 software (InVicro, USA).

### T4 measurement

Total T4 levels were measured by Mouse/Rat T4 Total ELISA kit (T4044T-100 Calbiotech) according to the manufacturer's instructions.

### T3 measurement

T3 levels were measured in 50 μl of plasma after extraction with chloroform-methanol 2:1 containing 1 mM propylthiouracil prior to measurement by highly sensitive radioimmunoassay, using an antibody produced by Gabriela Morreale de Escobar as described by in a supplement by Ferrara et al.[87]. The limit of detection was 5 ng/dL.

### TSH measurement

TSH was measured in 50 μl of plasma using a sensitive, heterologous, disequilibrium, double-antibody precipitation RIA. Due to the expected high concentrations of TSH in serum of RAI-treated mice, proper dilutions were made using TSH null mouse serum[88]. The limit of detection was 10 mU/L.

### Immunofluorescence staining

For immunofluorescence staining, cells cultured in monolayer or MTG-drop were fixed in 4% paraformaldehyde (PFA; Sigma) for 2 h at RT, washed three times in PBS, and blocked in 3% bovine serum albumin (BSA; Sigma), 5% horse serum (Invitrogen), and 0.3% Triton X-100 (Sigma) PBS solution for 30 min at room temperature. Primary and secondary antibodies were diluted in a PBS solution of 3% BSA, 1% horse serum, and 0.1% Triton X-100. Primary antibodies were incubated overnight at 4 °C, then washed three times and incubated with secondary antibodies for 2 h at room temperature. The nuclei were stained with Hoechst 33342 (Invitrogen). The slides were mounted with Glycergel (Dako). For paraffin embedding, in vitro organoids and grafted samples were fixed overnight at 4 °C in 4% PFA and kept in 70% ethanol at 4 °C for at least 24 h at 4 °C before embedding. Samples were then embedded in paraffin, sectioned (5 μm), mounted on glass slides, deparaffinized, and rehydrated. For histological analysis, sections were stained with hematoxylin and eosin (H&E) according to a routine protocol. For immunostaining, antigen retrieval was performed by incubating the sections for 10 min in the microwave (850 W)

in Sodium Citrate Buffer (10 mM Sodium Citrate, 0.05% Tween 20, pH 6.0). After cooling, the sections were rinsed with PBS and then blocked with 1% BSA and 10% horse serum PBS solution for 1 h at RT. Primary antibodies were diluted in the blocking solution and incubated overnight at 4 °C. The sections were rinsed three times in PBS and incubated with Hoechst 33342 (Invitrogen) and secondary antibodies diluted in blocking solution for 1 h at room temperature. Slides were mounted with Glycergel (Dako). Information on antibodies and sources are listed in Supplementary Table 3.

## Follicles size measurements
Images from grafted samples and human thyroid tissue were used for follicles size assessment using the Leica AF Lite Software using the bar scale tool. At least 150 follicles were analyzed for each condition. Data are shown as median (IQR).

## Assessment of T4-producing follicles proportion in vivo
Thyroid follicles were manually counted for the presence or absence of intra-lumenal T4 staining and the results are provided as percentage (%). In total, 480 follicles were counted using 20 immunofluorescence images of grafted tissue from four distinct mice samples. The follicles' structure was determined by the co-staining for NKX2-1, T4 and either E-CADHERIN or TG.

## Imaging
Fluorescence imaging was performed on a Zeiss LSM510 META confocal microscope, a Zeiss Axio Observer Z1 microscope with Axio-CamMR3 camera, and a Leica DMI6000 with DFC365FX camera. The images were acquired and generated using Zen software version 3.1 (Carl Zeiss) and LAS AF Lite software version 4.0 (Leica), respectively. Hematoxylin and eosin whole slide images were acquired using a NanoZoomer- SQ digital slide scanner C13140-01 (Hamamatsu) and images were generated using NDP.view2 software version 2.9.29 (Hamamatsu).

## Statistical analysis
Statistical significance between two groups was tested using the non-parametric unpaired Mann–Whitney $U$ test. Nonparametric Spearman test was used to calculate the correlation between two variables, presented as $r$ value. Data are presented as median (IQR). Differences were considered significant at $p < 0.05$. GraphPad Prism Software version 9 was used for most analyses (GraphPad Software). Data presented are from at least three independent experiments.

## Reporting summary
Further information on research design is available in the Nature Portfolio Reporting Summary linked to this article.

## Data availability
The data that support this study are available from the corresponding author upon request. Bulk RNA-seq and Single-cell RNA-seq data have been deposited in the NCBI Gene Expression Omnibus under accession number GSE181452 and GSE203085, respectively. In addition, data from bulk RNA-seq control samples (cAMP condition from day 37 and 45) have been deposited under GSE201558 accession number. scRNAseq data from human thyroid tissue was obtained from Human Cell Landscape (available at https://db.cngb.org/HCL/). The custom-built reference was based on the human reference genome GRCh38 (GenBank accession code GCA_000001405. 15; RefSeq accession code GCF_000001405). Source data are provided with this paper.

## Code availability
Custom computer script used to generate scRNAseq data are available upon request.

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

## Acknowledgements

The authors acknowledge Professor Andrew G Elefanty for kindly providing human ESC cell line (HES3-NKX2-1$^{WT/GFP}$). The ULB flow cytometry platform (C. Dubois), the ULB genomic core facility (F. Libert and A. Lefort), LiMIF platform for confocal microscopy (J.-M. Vanderwinden), Coraline De Maeseneire, Nicolas Passon and Christophe Van Heymbeek (CMMI) for their contribution for the in vivo studies. We also acknowledge Viviane de Maertelaer for the support with the statistical analysis. Schematic grafting diagram was created with BioRender.com. We acknowledge the funding agencies that supported this study. The Belgian National Fund for Scientific Research (FNRS) (PDR T.0140.14; PDR T.0230.18, CDR J.0068.22), the Fonds d'Encouragement à la Recherche de l'Université Libre de Bruxelles (FER-ULB) and the European Union's Horizon 2020 research and innovation program under grant agreement No. 825745 (S.C.). The Brazilian National Council for Scientific and Technological Development (CNPq; Brazil; M.R.). MISU funding from the FNRS (34772792 – SCHISM; S.P.S.). The Belgian Fondation contre le cancer (F/2020/1402; V.D.). The European Regional Development Fund (ERDF), the Walloon Region, the Fondation ULB, the Fonds Erasme and "Association Vinçotte Nuclear" (AVN) (G.D.). European Regional Development Fund (ERDF) and the Walloon Region (G.D.). FNRS (Chargé de Recherche, No.825745; M.R.). The National Institutes of Health (USA) (DK15070; B.F.F. and S.R.).

## Author contributions

M.R. and S.C. developed the project, designed the experiments and analyzed the data. D.F.K generated and selected the hESC-NKX2-1/PAX8 line. M.R., B.F.F., and O.M. performed most of the in vitro experiments and protocol set up. L.C. performed cell culture, maintenance and karyotype analysis. S.C., M.R., and G.D. executed in vivo studies. G.D., G.V.S., and S.G. acquired SPECT-CT images. X.H.L. measured T3 and TSH in plasma. M.R., B.F.F., and H.L. analyzed RNA expression and performed IF. M.R. and P.G. obtained confocal images. M.R. and H.L. performed bulk RNA-Sequencing and analyzed the results. M.R., S.P.S., and B.F.F performed the single-cell RNA-Sequencing. A.T., V.D., B.F.F. S.P.S., S.E.E., and H.L. performed the bioinformatics analysis. S.C. and M.R. wrote the first draft and S.P.S. and S.R. edited the manuscript. S.C. and S.R. acquired funding for the project. All authors contributed to the article and approved the submitted version.

## Competing interests

S.C. and M.R. have filed EU patent regarding the hESC-thyroid derived organoids. All other authors declare no competing interests.
