## [Peer Review File · Nature Communications]

Transplantable human thyroid organoids generated from embryonic stem cells to rescue hypothyroidismREVIEWER COMMENTS

Reviewer #1 (Remarks to the Author):

Romitti et al present interesting progress on developing the thyroid organoid system from genetically modified human pluripotent stem cells. Transient overexpression of NKX2-1 and PAX8 expedite differentiation into thyroid hormone secreting follicular-like organoids. Given that thyroid organoid formation in humans is only successful from primary tissue, this PSC-based approach represents a major advance in the field. However, there are a substantial lack of experimental evidence to support the authors' conclusion mainly due to a lack of adequate sample size, appropriate control, and quantitative description of the results. My major concerns follow:

1. Comparator primary tissues

Throughout the manuscript, not primary tissue-derived samples were compared to thyroid organoids, prohibiting the fair assessment of maturity levels and its differentiation stage. It is increasingly recognized as important (almost imperative) to benchmark comparative-stage derived primary tissues. This comparison includes evaluation of gene expression level, spatial relationship, and functional output (secretion level). Additionally, it would be also informative to discuss their data compared to published literature of primary tissue-derived organoids.

2. Reproducibility and validation of scRNAseq

scRNAseq is certainly an informative tool to analyze the compositional profile of organoids however, it is only tested in one timepoint (day45) and one small number of samples (1874 cells), raising significant concern for reproducibility and robustness. How much extent cell compositions and their differentiation status are preserved across the different batches? It is highly encouraged to perform additional sample sets ideally with primary control.

Additionally, there is a substantial lack of scRNAseq validation to evaluate the significance of the findings. For example, the author's protocol derives a significant number of off-target cells by scRNAseq but is never evaluated in further figures. These need to be validated in later figures to determine their real existence, frequency, and spatial relationship to follicular tissues.

3. Quantitative assessment on the efficiency of hormone-secreting organoids in vitro and in vivo

One major drawback is the lack of a quantitative description of the authors' system. This is persistently problematic in all of the Figures. I wonder 1. the efficiency/yield of the hormone-secreting organoids, 2. what types of off-target cells are formed and again how much (as noted in #2), and 3. the level of hormone production capability in vivo (normalized by single-cell) and their discussion to normal adult follicular cells.

4. Lack of discussion with recent progress around thyroid organoids

Related to the #1 comment, the below literatures are recent examples of primary derived thyroid organoid system that was not mentioned nor discussed in the manuscript. It is highly encouraged to discuss pros/cons from multiple angles (differentiation levels, sample accessibility, transgene needs, immunogenicity, etc...).

Jelte van der Vaart, et al. Adult mouse and human organoids derived from thyroid follicular cells and modeling of Graves' hyperthyroidism. Proceedings of the National Academy of

Sciences 118 (51) e2117017118, (2021).

V. M. L. Ogundipe et al., Generation and differentiation of adult tissue-derived human thyroid organoids. *Stem Cell Reports* 16, 913–925 (2021).

Y. Saito et al., Development of a functional thyroid model based on an organoid culture system. *Biochem. Biophys. Res. Commun.* 497, 783–789 (2018).

A. Khoruzhenko et al., Functional model of rat thyroid follicles cultured in Matrigel. *Endocr. Connect.* 10, 570–578 (2021).

Reviewer #2 (Remarks to the Author):

Romitti et al perform T4-producing human thyrocytes from pluripotent stem cells. It seemed very difficult to achieve. The article lacks clarity (especially the first parts) and discussion. It's a shame because the work could be much more interesting. The part of single cell characterization of human thyroid organoid is very interesting.

There are several problems with the methodology (lack of controls), data and discussion, errors in the text. Whilst on a subject of potential interest, the paper is weak in several aspects.

Recommended points to address:

How do you explain difference of protocol between human and mice? Discuss please
Why it seems so difficult to obtain T4-producing human thyroid follicles?
Could you better explain and discuss each step of the protocol, and add controls?

Lines 38-39: Affirmation to be justified, publications? Too much affirmative, not real in clinical practice. It is not a strong argument. In comparison with immunosuppressive treatment needed for a transplant. Great interest for development biology, understanding... but for a replacement treatment for Levothyrox.

// Lines 24-25: This argument is not correct in abstract.

Line 79: Fig 1a: Add day of culture corresponding at each stick like Day 0 and Day 58

Line 80: Supp Fig1d: indicate stage of culture

Supp Fig 1e: line 804: replace l with e, add h

Line 90: Add co-immunostaining of NKX2-1 and PAX8

“large proportion”: could you give a percentage?

Lines 99-101: expression of endogenous NKX2-1: follows the PAX8 curve? And FOXE1, HHEX?

Could you conclude only with PAX8?

In Fig 1c: NKX2-1 seems to be very low in comparison with PAX8, why?

Line 101: direct addition of cAMP at D9 is not effective? Did you try without the proliferation

week?

Line 105: Add control: without cAMP

Could you show proliferation between D9 and D16? And difference with D20 after 2 weeks of cAMP?

Why control cells (stem cells) do not proliferate (supp fig 2a)?

Supp Fig 2b: 2 columns per stage?

Lines 109-110: why NKX2-1 increases and FOXE1 and PAX8 decrease between D16 and D23/30?

Fig 1c: no control on heatmap? And on all heatmaps

Controls are needed for each step, +/- dox, cAMP, TSH.....

Lines 115-117: PAX8 in the text and NKX2-1 in Supp 2d. Did you perform GFP co-immunostaining to check that TG-cells are the NKX2-1-GFP-cells?

Line 122: Could you explain more precisely the use of Dexamethasone?

Fig 1c: Could you give the quantification between D38 and D45? DUOXA2, TPO and TSHR decrease on heatmap. Supp 2e: strong increase between D45 +/-Dox but between D36 and D45, +/- Dox, +/- hTSH +/- Dexa, +/- SB

Controls are needed.

Line 130: supp Fig 2g: co-immunostaining with TG, TG-I, T4?

PAX8 in the text, NKX2-1 in figure?

Line 132: 25% of NKX2-1 cells, is it very efficiency? In comparison with other protocols?

Discuss please

What is the decrease in proliferation at D47 due to: differentiation, apoptosis, etc.

Line 139: it's not clear in the Fig 1c

Line 142: did you perform T4 immunostaining?

Line 147: why enrichment of 60% of NKX2-1-GFP+ cells? How did you choose this %?

Line 164: NKX2-1 GFP+ cells (60%) correspond to thyroid progenitors, immature thyrocytes and mature thyrocytes, airway cells

NKX2-1 neg cells (40%) correspond to fibroblasts, cardiovascular and endoderm epithelial cells? Are there other cells?

Line 169: in which figure?

Line 170: Did you check/validate crosstalk, that you observe with CellPhone-DB, between thyroid cells and the other cells? Did you perform culture without cardiovascular cells for example?

Line 203: quantifications of T4, TG and TPO are needed, comparison between D45 and D58?

Did you perform T4 assay in culture medium?

Did you analyze proliferation from D47 to D58? And apoptosis?

How long do T4-producing cells survive in culture in your model?

How many stem cells are needed to reproduce a functional human thyroid (11/12weeks)?
Same if it is a proof of concept. Did you compare T4-immunostaining? Quantification? (Fig 2d and compare with human thyroid tissue)

Line 235: Did you perform NIS immunostaining on graft? And analyze of airway, cardiovascular cells... by immunostaining on graft?

Line 236: human tissue at each stage?

Line 251: Did you perform TSH assay to validate the rescue?

Line 263: Could you compare your protocol and your results with those of Kurmanns? (human cells)

Line 528: Could you precise "blocking solution"

Line 549: Supp table 3? There is no Table 3.

Line 676: What is MTG? explain please

Mann and Whitney test is more appropriate than t-test

References to be checked

7,8: incorrect references

15: Ogundipe et al.

...

**Response to Reviewers**

**Reviewer #1 (Remarks to the Author):**

Romitti et al present interesting progress on developing the thyroid organoid system
from genetically modified human pluripotent stem cells. Transient overexpression of
NKX2-1 and PAX8 expedite differentiation into thyroid hormone secreting follicular-
like organoids. Given that thyroid organoid formation in humans is only successful from
primary tissue, this PSC-based approach represents a major advance in the field.
However, there are a substantial lack of experimental evidence to support the authors'
conclusion mainly due to a lack of adequate sample size, appropriate control, and
quantitative description of the results. My major concerns follow:

1. Comparator primary tissues

Throughout the manuscript, not primary tissue-derived samples were compared to
thyroid organoids, prohibiting the fair assessment of maturity levels and its
differentiation stage. It is increasingly recognized as important (almost imperative) to
benchmark comparative-stage derived primary tissues. This comparison includes
evaluation of gene expression level, spatial relationship, and functional output
(secretion level). Additionally, it would be also informative to discuss their data
compared to published literature of primary tissue-derived organoids.

**R. Dear reviewer, thank you for your suggestions. As required, we compared our**
**results to primary thyroid tissue as described below:**

**qRT-PCR was used to compare our thyroid organoids (D58) and derived thyroid tissue**
**after transplantation (Graft) to human adult thyroid tissue. Fig. 4e. Lines 321-325.**

**Immunostaining was used to compare the resulted thyroid tissue after graft to the**
**human adult thyroid sample. The cell organization and protein expression show that**
**organoids-derived tissue resembles the architecture of adult human thyroid, however**
**the follicles size range is lower in graft compared to tissue. It might indicate that our**

organoids might represent an early stage of fetal thyroid development (Szinnai, 2007).
Fig. 4d and Supplementary Fig. 6e.

Despite the indications that at day 58 or after 5 weeks of transplantation, our organoids
resemble fetal thyroid, we aimed to compare our data to different stages of thyroid
development. Dom, et al, 2021, profiled the thyroids of human fetuses. We hoped
these data would be a good source of comparison. Unfortunately, the dramatic
technical variation between our RNAseq data and the Affymetrix arrays of Dom et al.
and the limited amount of data available for batch effect correction precluded any
meaning full comparison.

Furthermore, we have also contacted the authors of the article “Modeling Human
Thyroid Development by Fetal Tissue-Derived Organoid Culture, Adv. Sci, 2022”
aiming to compare their fetal thyroid organoids scRNAseq data to ours, but we didn’t
obtain any reply and since the data is not publicly available, we could not perform the
analysis.

Regarding comparison with adult thyroids, The Human Cell Landscape
(<https://db.cngb.org/HCL/>) includes two ‘normal’ single cell RNAseq thyroid profiles.
We processed both with our pipeline. One of them included a high percentage of
lymphocytes suggesting a thyroiditis. The other one had reasonable cell types
proportions. The Human Cell Landscape used the microwell-seq and lower per-cell
read counts, while we used 10X Chromium platform. After applying a batch effect
correction algorithm to resolve these technical differences, we found a significant
overlap between the cells of thyroid organoids and those of human thyroid tissue. The
analysis is presented at Fig. 3e and f.

2. Reproducibility and validation of scRNAseq
scRNAseq is certainly an informative tool to analyze the compositional profile of
organoids however, it is only tested in one timepoint (day45) and one small number of
samples (1874 cells), raising significant concern for reproducibility and robustness.
How much extent cell compositions and their differentiation status are preserved
across the different batches? It is highly encouraged to perform additional sample sets
ideally with primary control.

R: Dear reviewer, we agree that scRNAseq at another time point, and from human
thyroid tissue would bring new information to the article and reinforce the data from
64 day 45. For that we performed scRNAseq at day 58, when the system is functional,
and we also used the publicly available human cell landscape dataset from adult
thyroid tissue.

At Fig. 3a-c and Supplementary Fig. 3b, you can find the characterization of the
organoids at day 58 as well as the thyroid clusters identified. In addition, we performed
integration analysis initially with organoids from D45 and D58, showing strong overlap
between the time points (Fig. 3d); Secondly, we performed integration analysis among
the thyroid organoids datasets and the adult thyroid human cell landscape dataset (we
have excluded the sample in which most of cells were immune cells), which shows an
overlap between the hESC-derived thyrocytes and the adult thyroid cells (Fig. 3f).

As mentioned above the scRNAseq data from “Modeling Human Thyroid Development
by Fetal Tissue-Derived Organoid Culture, Adv. Sci, 2022” article is not publicly
available, so we could not perform the comparison analysis.

Additionally, there is a substantial lack of scRNAseq validation to evaluate the
significance of the findings. For example, the author's protocol derives a significant
number of off-target cells by scRNAseq but is never evaluated in further figures. These
need to be validated in later figures to determine their real existence, frequency, and
spatial relationship to follicular tissues.

R. We do share this concern as our study suggests that off-target cells are important
to effectively generate organoids that release thyroid hormones in the bloodstream.
As scRNAseq only provide a quantitatively biased picture of cell types composition
and does not inform spatial structures, we addressed this point with specific targeted
approaches. We first characterized each non-thyroid cluster by specific markers at
mRNA levels and performed IF to confirm the existence of such cell types and our
classification (Supplementary Fig. 4a-m). Furthermore, as predicted by connectome,
mesoderm might interact with thyroid cells through many distinct pathways. Hence,
we performed IF to determine the special location of such population and the proximity
with thyroid cells. Interestingly, we noticed that fibroblasts and cardiovascular cells

were in close contact with thyroid organoids, reinforcing the possible interaction
between them (Supplementary Fig. 5a-f). Even if at less extent, airway and epithelial
cells also seem to be close to thyroid cells as demonstrated at Supplementary Fig. 4k
and m.

Regarding the frequency of the non-thyroid cell types, due to technical issues we did
not have all markers (Ab) suitable for Flow cytometry quantitative analysis. In addition,
since we demonstrate by scRNAseq that compared to day 45, at day 58 the cluster of
fibroblasts has almost completely disappeared. The only conclusion we are able to
draw at this point is that the proportions of non-target cells can vary during the protocol.
Quantitative details will be addressed in future studies.

3. Quantitative assessment on the efficiency of hormone-secreting organoids *in vitro*
and *in vivo*
One major drawback is the lack of a quantitative description of the authors' system.
This is persistently problematic in all of the Figures. I wonder 1. the efficiency/yield of
the hormone-secreting organoids, 2. what types of off-target cells are formed and
again how much (as noted in #2), and 3. the level of hormone production capability *in*
*vivo* (normalized by single-cell) and their discussion to normal adult follicular cells.

R: Dear reviewer, as required we included the controls in the figures and included a
new Supplementary Fig. 2 where we show the controls for each change of the protocol
and how the gene expression changed according to the different treatments. We also
include the RNAseq heatmaps showing the expression profile using the new treatment
compared to cAMP control condition. For the time course bulk RNAseq (Fig. 1g), we
have not included -Dox condition as negative control, since the purpose is not to
compare it to an uninduced condition, but among the different stages of thyroid
development *in vitro*.

In order to measure TH yield *in vitro* we performed T4 and T3 dosage using the ELISA
kit and we could not detect clear values, which might be due to the limit of detection
of the techniques. In addition, following your suggestion, we tried to estimate the
percentage of functional follicles. Because we used paraffin sections for T4
immunostaining (5 μ m), counting follicles may not be appropriate because the same
follicle may be present on different sections.

The levels of THs *in vivo* were measured and compared to non-transplanted control
mice. Unfortunately, we can't compare the THs levels obtained from *in vivo*
transplantation to patients' levels, since the number of follicles, maturation degree and
TSH response are highly distinct. In addition, measurement of T4 production
(plasmatic level)/cell would require complete dissociation of all transplanted tissue and
counting of thyrocytes by flow cytometry. This could give us a T4 level/cell that would
be of little value anyway because of TSH regulation.

4. Lack of discussion with recent progress around thyroid organoids
Related to the #1 comment, the below literatures are recent examples of primary
derived thyroid organoid system that was not mentioned nor discussed in the
manuscript. It is highly encouraged to discuss pros/cons from multiple angles
(differentiation levels, sample accessibility, transgene needs, immunogenicity, etc...).

Jelte van der Vaart, et al. Adult mouse and human organoids derived from thyroid
follicular cells and modeling of Graves' hyperthyroidism. *Proceedings of the National*
*Academy of Sciences* 118 (51) e2117017118, (2021).

142 V. M. L. Ogundipe et al., Generation and differentiation of adult tissue-derived human
thyroid organoids. *Stem Cell Reports* 16, 913–925 (2021).

Y. Saito et al., Development of a functional thyroid model based on an organoid culture
system. *Biochem. Biophys. Res. Commun.* 497, 783–789 (2018).

148 A. Khoruzhenko et al., Functional model of rat thyroid follicles cultured in Matrigel.
*Endocr. Connect.* 10, 570–578 (2021).

R: Thank you for pointing it out. Indeed, those are very interestingly articles, and some
were not mentioned before since they were published after the first submission of the
paper. We have included a paragraph in the introduction (Page 3, L.54-67) and we
have discussed the suggested points with pros and cons of this model system
compared to ESCs-derived models. (Page 16, L.373-381; Page 17-18, L. 421-428).

**Reviewer #2 (Remarks to the Author):**

Romitti et al perform T4-producing human thyrocytes from pluripotent stem cells. It
seemed very difficult to achieve. The article lacks clarity (especially the first parts) and
discussion. It's a shame because the work could be much more interesting. The part
of single cell characterization of human thyroid organoid is very interesting.

There are several problems with the methodology (lack of controls), data and
discussion, errors in the text. Whilst on a subject of potential interest, the paper is weak
in several aspects.

Recommended points to address:

How do you explain difference of protocol between human and mice? Discuss please

R: Dear reviewer, the difference between both protocols might start by the fact that
mESC and hESC are different at basal state. As you can find in the discussion (Page
15, L.360-372) mESCs are considered as naïve while hESC are considered as primed
stem cells, it means that naïve mESCs can efficiently contribute to blastocyst
chimeras, whereas primed SCs have the potential to differentiate into primordial germ
cells *in vitro*. Moreover, not only pluripotency is not the same between the two stem
cells, but they also differ in timing of gene expression, levels and epigenetic
modifications. For example, studies have shown that neuron and sensory organ
develop differently from mESCs and hESCs (Gabdouline, 2015). Despite the
progresses made in the last years to induce a “more naïve” hESC culture method (as
used here by Stem Flex medium hESC culture), we cannot assume that both cells are
at the same stage of development and can differentiate similarly. The second aspect
is that for human the endoderm step is crucial, since spontaneously the hESC do not
generate substantial amount of endoderm cells. On the contrary of mouse model,
human thyrocytes cannot mature completely using only cAMP treatment, and the
addition of TSH is essential for the final maturation, which might indicate differences
in gene expression regulation between the two systems. Also, the timing of
development seems to follow the one observed *in vivo* probably the reason why the
human protocol takes around 60 days while mouse 21 days.

Why it seems so difficult to obtain T4-producing human thyroid follicles?

R: Based on our findings and from previous studies (Kurmann, 2015; Ma 2015, 2017,
2020; Serra, 2017) efficient generation of thyroid from pluripotent SCs is a challenge
and the protocols require several steps in order to generate thyroid cells. Most of the
studies show that this stage can be reached (thyrocytes derivation), but proper 3D
organization with thyroid hormone synthesis *in vitro* and *in vivo* was not achieved from
hPSCs. Many factors can lead to these difficulties such as, efficient endoderm-
induction and thyroid specification, 3D environment, cell culture media (most of the
studies don't use FBS, but defined supplementation, which can lead to the absence of
essential growth factors and other molecules) and the purified thyroid population, used
in most of the models can constitute a disadvantage since other non-thyroid cells can
play a role on thyroid development and maturation. An interesting example is shown
by Ogundipe, et al, in which adult tissue-derived thyroid organoids were transplanted
and needed 26 weeks to form functional tissue *in vivo*. Indicating that such pure
population might be inhibitory for thyroid formation and function, even when starting
from a mature and functional source of material.

Could you better explain and discuss each step of the protocol, and add controls?

R: Dear reviewer, we have included new nomenclature and experiments to provide
more information about the different stages of thyroid organoids development *in vitro*.

- At Fig. 1a, we renamed the stage from day 9-16 to thyroid fate commitment
instead of thyroid proliferation since the idea of this week without
supplementation was to allow the thyroid endogenous machinery activation.
And shown at new Fig. 1c, we performed flow cytometry for co-expressing
NKX2-1GFP and PAX8 cells and observed that around half of the cells that co-
expressed the two TF at day 9 keep the endogenous activation and day 16,
being the endogenous "progenitors". We also replaced the previous KI67
experiment by BrdU assay, and we observed that during this period of time,
cells uptake less BrdU, which reinforce the balance between differentiation and
proliferation (Fig. 1e). Of note, BrdU control, were cells treated with BrdU
without the Ab. Other controls were used for the verification of the analysis,

including -Dox condition (treated with BrdU and Ab) and cells not treated with
BrdU and incubated with the Ab.

- We also characterized by BrDU assay that from day 16 to 30 (cAMP treatment),
thyroid cells expansion occurs with a clear increase in the proportion of
proliferating cells expressing BrdU (Fig. 1e).

- We also included the Supplementary Fig. 2a-m, where we show by qRT-PCR
effect of the switch from cAMP supplementation to hrTSH +/-or Dexa +/-or
Sb431542 on thyroid gene expression (at day 38 and 45), showing all the
controls: -Dox, cAMP, cAMP+Dexa, hrTSH, hrTSH+Dexa, hrTSH+SB. In
addition, bulk RNAseq data was used to compare the inflammation, TGF-beta
and thyroid markers, between cAMP condition (control) and the new treatment.

- Immunostaining for TG was included starting from D30, showing the
progression of maturation after the medium supplementation switch (Fig. 1i).

- Day 45 was also now characterized by IF for the main thyroid protein markers
and to show the histological organization of the cells at this time point (Fig. 2a-
238 d).

- In addition, -Dox control was included in all qRT-PCR graphs as a column not
only as fold to -Dox as previously mentioned (Fig. 1c, d, h; Fig. 3g).

Lines 38-39: Affirmation to be justified, publications? Too much affirmative, not real in
clinical practice. It is not a strong argument. In comparison with immunosuppressive
treatment needed for a transplant. Great interest for development biology,
understanding... but for a replacement treatment for Levothyrox.

R: We modified this paragraph, and we added some references supporting this
discussion.

250 // Lines 24-25: This argument is not correct in abstract.

R: We deleted this sentence in this abstract because it seemed too ambitious. We
have replaced it with a few words that still leave room for an ambitious vision for the
medicine of the future.

One of the long-term therapeutic perspectives is, of course, to solve the problem of
immune tolerance. However, this problem, which is at the heart of regenerative
medicine, is already largely explored through approaches using patient-derived iPSC
or embryonic stem cells with universalization of the HLA system (*Riolobos L, Hirata*
*RK, Turtle CJ, Wang PR, Gornalusse GG, Zavajlevski M, Riddell SR, Russell DW.*
*HLA engineering of human pluripotent stem cells. Mol Ther. 2013 Jun;21(6):1232-41.*)
It therefore seems ambitious but scientifically relevant to investigate the same issues
in the context of thyroid medicine.

Line 79: Fig 1a: Add day of culture corresponding at each stick like Day 0 and Day 58

R: Thank you for your observation, days were included in the Fig.1a scheme of the
differentiation protocol.

Line 80: Supp Fig1d: indicate stage of culture

R: Dear reviewer, as suggested, we also included the stage of the culture to the figure
legend. It was already described in the methods “**Generation of tetracycline-induced**
**hESC line**” (Page 18, L. 460-462).

Supp Fig 1e: line 804: replace l with e, add h

R: Thank you for pointing it out. Figure, legend, and text were modified accordingly.

Line 90: Add co-immunostaining of NKX2-1 and PAX8
“large proportion”: could you give a percentage?

R: Dear reviewer, the proportion of NKX2-1 and PAX8 co-expressing cells at day 9
and 16 is shown in the new Fig. 1c.

Lines 99-101: expression of endogenous NKX2-1: follows the PAX8 curve? And
FOXE1, HHEX? Could you conclude only with PAX8?

R: Indeed, this is an interesting aspect and we explored how could we analyze NKX2-
1 expression by qRT-PCR, since in our Knock-in line GFP is replacing the exon 2 of
NKX2-1 gene (which has only 3 exons). We have tested several combinations of
primers, and due to the specificity (exon-exon junction design) we could not effectively
measure the mRNA. So, we decided to exclude the NKX2-1 gene expression from our
qRT-PCR analyses (Page 5, L104-106). On the other hand, we observed by IF that all
cells expressing PAX8 also expressed NKX2-1, which is not the case in the contrary
situation (Fig. 1b). So, we believe that the endogenous curve for PAX8 reflects the
thyroid fate commitment. In addition, as suggested, FOXE1 and HHEX expression
curve was included in the Supplementary Fig. 1g. Please note that HHEX expression
is not following the thyroid induction, since it is a classical endoderm marker, which in
our case is induced by AA in our protocol.

In Fig 1c: NKX2-1 seems to be very low in comparison with PAX8, why?

R: Thank you for the observation. There is indeed a technical effect pertaining to the
NKX2-1-GFP construct. As mentioned above in our NKX2-1-GFP line, one of the
alleles of NKX2-1 gene has GFP replacing exon 2, and since it interferes in mRNA
expression analysis it likely impacts the number of reads aligned on NKX2-1 in bulk
RNAseq which covers the entire transcript thanks to the random priming protocol.

On the other hand, as demonstrated in the Fig. 2g, Fig. 3c and Supplementary Fig. 3
a-b, scRNAseq did not reveal any clear difference in the levels, very likely since in this
approach only the last 100-150bp from the 3' UTR are sequenced.

To the best of our knowledge, no study provides a strong reference to compare PAX8
and NKX2-1 levels during thyroid development *in vitro* or *in vivo*. We opted to show
the data as neutrally as possible, and refrain to emphasize any dataset.

Line 101: direct addition of cAMP at D9 is not effective? Did you try without the
proliferation week?

R: Dear reviewer, indeed that is an interesting point and we have renamed this step
(from day 9-16) to thyroid fate commitment, since it is when the Dox-induced cells
should start to express the endogenous thyroid machinery and activate the thyroid
differentiation program, which does not reflect the proliferation stage of the protocol.
So, when we treated the cells with cAMP at day 9 we observed increase in thyroid
population proliferation measured by BrdU uptake at day 16 compared to the untreated
situation (+cAMP 8.9% and -cAMP 5,9%). This reflected the % of NKX2-1^{GFP} with
around 50% more cells in the cAMP treated condition compared to the control (+cAMP
4.6% and -cAMP 3.0%). Also, we saw by qRT-PCR that PAX8 and FOXE1 levels are
slightly higher while TG is similar and TSHR is lower. Due to the higher number of
cells in the +cAMP condition we normalized once more the data using PAX8 levels
and we saw that the levels of the differentiation markers TSHR and TG are lower
compared to the untreated situation, indicating that the balance between of
proliferation and differentiation is important promoting thyroid commitment fate. Data
is show in the graphs bellow.

Line 105: Add control: without cAMP

R: Dear reviewer, the new version of the paper includes a BrdU experiment instead of
KI67 assessment. For that we used several different controls, such as, -Dox (to set
the gates), treated (all treatments) cells incubated of not with BrdU and/or with and
without anti-BrdU antibody. As the goal here was to determine the curve of proliferative

cells over time based on NKX2-1/GFP co-staining with BrdU the control is the treated
(BrdU) cells without anti-BrdU.

Could you show proliferation between D9 and D16? And difference with D20 after 2
338 weeks of cAMP?

R: As required, in the Fig. 1e we demonstrate the profile of BrdU+ among NKX2-1
expressing cells during this period and compared to the other time points/

Why control cells (stem cells) do not proliferate (supp fig 2a)?

R: Since for proliferation assay, we analyze BrdU positivity among NKX2-1 expressing
uninduced cells cannot be used as control (which in this case would be -Dox condition,
which comprises many differentiated cell types, not being at a stem cell state
anymore), so as control cells treated with BrdU solution and not stained were used as
control for the experiment. Please see Fig. 1f where we demonstrate that -Dox
condition has extremely low % of spontaneously differentiated NKX2-1 cells.

Supp Fig 2b: 2 columns per stage?

R: Dear reviewer, we have performed the experiment in duplicates, for which one we
have used at least 6 wells pooled together in order to overcome the possible variability
among wells (information is included in the “RNA-seq and analysis of bulk samples”
session described in Methods). The goal here was to see the changes in thyroid genes
expression overtime during the differentiation protocol, using the previous time point
as control, since we showed by qRT-PCR at several time points the thyroid gene
expression compared controls (Fig. 1. d; h; Fig. 3g; Supplementary Fig. 2 c-l). Also,
for the RNAseq data we included cAMP as control for TGF-beta, inflammation and
thyroid markers comparing to the medium changes from day 30 onwards
(Supplementary Fig. 2a, b, m).

Lines 109-110: why NKX2-1 increases and FOXE1 and PAX8 decrease between D16
and D23/30?

R: At day 16 we could expect still to have an effect of the exogenous induction of
NKX2-1 and PAX8, which could explain higher levels of PAX8 and FOXE1. NKX2-1
levels might be affected since it is a knock-in knock-out in our cell line.

Fig 1c: no control on heatmap? And on all heatmaps. Controls are needed for each
step, +/- dox, cAMP, TSH.....

R: As explained above, the meaning of the bulk RNAseq here is not to show induction
of differentiation but see the changes in gene expression overtime during the
differentiation protocol and the comparison it is made among time points. Besides that,
we included RNAseq data using cAMP as control to see changes in TGF-beta,
inflammation and thyroid markers comparing to the medium changes from day 30
onwards (Supplementary Fig. 2a, b, m).

Lines 115-117: PAX8 in the text and NKX2-1 in Supp 2d. Did you perform GFP co-
immunostaining to check that TG-cells are the NKX2-1-GFP-cells?

R: Dear reviewer, we apologize for the mistake, and it has been corrected, thank you
for pointing it out. We have observed by IF that all the TG-expressing cells are NKX2-
1/GFP positive, however not all NKX2-1/GFP cells are TG positive since we can have
airway and goblet cells.

Line 122: Could you explain more precisely the use of Dexamethasone?

R: Since studies have shown that high inflammation impact thyroid genes expression
(see list of references bellow) we evaluated the levels of inflammation by bulk RNAseq
at days 38 and 45 when cells were treated only with cAMP (Supplementary Fig. 2a).
Also, qRT-PCR data showed the dexamethasone effect on NIS (D38) and the addition
to SB increasing significantly the levels of most of the genes including NIS, TG and
TPO at day 45 compared to distinct control conditions (Supplementary Fig. 2c-l).

References:

Mori, K., Mori, M., Stone, S., Braverman, L. & DeVito, W. Increased expression of
tumor necrosis factor-alpha and decreased expression of thyroglobulin and thyroid
peroxidase mRNA levels in the thyroids of iodide-treated BB/Wor rats. *European*
*Journal of Endocrinology* 139, (1998).

Dohán, O. et al. The sodium/iodide symporter (NIS): Characterization, regulation, and
medical significance. *Endocrine Reviews* vol. 24 48–77 (2003).

Faria, M. et al. TNF α -mediated activation of NF- κ B downregulates sodium-iodide
symporter expression in thyroid cells. *PloS one* 15, e0228794–e0228794 (2020).

Fig 1c: Could you give the quantification between D38 and D45? DUOXA2, TPO and
TSHR decrease on heatmap. Supp 2e: strong increase between D45 +/-Dox but
between D36 and D45, +/- Dox, +/- hTSH +/- Dexa, +/- SB
Controls are needed.

R: Dear reviewer, as required we have included the qRT-PCR data from days 38 and
45 for most of the thyroid genes including several controls, see Supplementary Fig. 2.
c-l. In addition, since our designed primers for quantification of DUOXA2 didn't provide
acceptable curve and trustful data, bellow you can find the data concerning DUOXA2
gene expression (C.P.M.) from bulk RNAseq at both time points, compared to the
cAMP control.

Line 130: supp Fig 2g: co-immunostaining with TG, TG-I, T4?

R: Dear reviewer, new confocal images from day 45 are included in the new Fig. 2 a-
414 d.

PAX8 in the text, NKX2-1 in figure?

R: Thank you for spotting the mistake, the figure was corrected accordingly.

Line 132: 25% of NKX2-1 cells, is it very efficiency? In comparison with other
protocols? Discuss please

R: Dear reviewer, since obtained around 25% of the cells at the end of the protocol
which are NKX2-1+ and considering that our model is multicellular and that the other
cells are expected to be more proliferative than thyroid, the efficiency is significant
considering human models. Our thyroid mouse model and from other groups show
efficiency ranging from 24-60% of the cells (Posabella, et al, Frontiers in Endoc, 2021),
showing that our human model is close to this range of efficiency. However, proper
comparison to all protocols of human thyroid-derived (PSCs models) organoids is not
possible since the studies don't describe clearly the efficiency of thyroid generation.
Arauchi, et al, (Frontiers in Endoc, 2017) describes an efficiency of 16% of the thyroid
cells generation at day 20 of their protocol. Ma, et al, (Frontiers in Endoc, 2020) shows
that after Endoderm and specification induction 34% of the cells co-expressed NKX2-
1 and PAX8, compared to our protocol in which we observed 80% of the cells co-
expressing both TFs at day 9. In addition, they show that at day 21 of the protocol,
TSHR/NIS co-expressing cells were around 58% and after sorting could reach 97% of
purification.

What is the decrease in proliferation at D47 due to: differentiation, apoptosis, etc.

R: Dear reviewer, since we decided to change the assay for measuring precisely the
cells proliferating, at the new Fig. 1e, we show that proliferation is clearly reduced from
440 day 37 compared to day 30, which follows the change of media supplementation from
441 cAMP to hrTSH+Dexa. It is clear that cAMP is promoting proliferation in our system
as you can observe by the improve in % of proliferating cells from day 16 onwards. In
addition, we can observe that the media change at day 30 resulted in increase in
thyroid differentiation and maturation, shown at Fig. 1g, suggesting the balance
between differentiation and proliferation in our system. Also, it is important to notice,
that despite the decrease in the proliferation rate from D37, the % of NKX2-1 cells is
stable (Fig. 1f). Furthermore, to address the possible process of apoptosis, we

performed Caspase3 staining in our organoids (D58) and we did not observe any
expression among thyroid cells (see example below).

Line 139: it's not clear in the Fig 1c

R: The sentence was rewritten adequately (Page. 8, L.178). At Fig. 1d we
demonstrated that at day 9, *FOXE1*, *TSHR* and *TG* can be detected by qRT-PCR,
even if at lower levels compared to *NKX2-1* and *PAX8*. While by bulk RNAseq and
qRT-PCR (Fig. 1g and Supplementary Fig 2. c-l) we demonstrate that significant
expression of those genes and other maturation markers is observed from day 37
onwards.

Line 142: did you perform T4 immunostaining?

R: To better characterize the cells at day 45, you can find at the new Fig. 2 d the co-
staining for TG-I and T4 among NKX2-1 cells.

Line 147: why enrichment of 60% of NKX2-1-GFP+ cells? How did you choose this
465 %?

R: Dear reviewer, we enriched the thyroid population aiming to obtain enough thyroid
cells for characterization, since 25% GFP could result in very diluted distinct thyroid
population. Also, the aim was only identifying the other cells, not to estimate
proportion.

Line 164: NKX2-1 GFP+ cells (60%) correspond to thyroid progenitors, immature
thyrocytes and mature thyrocytes, airway cells NKX2-1 neg cells (40%) correspond to
fibroblasts, cardiovascular and endoderm epithelial cells? Are there other cells?

R: Dear reviewer, please see Fig. 2e-f and Fig. 3a-b. Since we performed a new
scRNAseq at day 58 we reanalyze our data set and reclassified the clusters, according
to the expression profile. However, we cannot exclude the presence of other cells in
very low proportion, such as fibroblasts at day 58. In addition, you can find the
characterization by immunostaining of those non-thyroid cells at Supplementary Fig.
4j-m and Supplementary Fig. 5c and f.

Line 169: in which figure?

R: Dear reviewer, this data is based in the cell counting from scRNAseq data. From
the total of NKX2-1 cells, more than 75% are also PAX8, this quantification is not
shown as graph, since the UMAPs from Fig. 2g and Fig. 3c show the clusters
expressing both TFs.

Line 170: Did you check/validate crosstalk, that you observe with CellPhone-DB,
between thyroid cells and the other cells? Did you perform culture without
cardiovascular cells for example?

R: Dear reviewer, in our culture condition, sorted GFP+ cells (day 23) which were
cultured alone did not survive. To show special distribution of thyroid cells and
mesodermal/cardiovascular cells, we used immunostaining which shows that both
populations are in close contact within our organoids (Supplementary Fig. 5c and f).

In addition, TGF-beta inhibition resulted in improvement of maturation as
demonstrated by several analysis within the paper. Concerning the role of
cardiovascular cells during thyroid development, we have a project ongoing aiming to
understand such relationship. For this reviewing, due to the complexity of the
experiment we could not perform it on time. For example, we observed that
dissociation of thyroid cells clusters, even at early stages, interfere in the thyroid 3D
generation, so removal of cardiovascular cells gets tricky.

Line 203: quantifications of T4, TG and TPO are needed, comparison between D45
and D58?

R: Dear reviewer, following your request we also performed immunostaining, qRT-
PCR and scRNAseq at day 58, in order to determine the differences in gene
expression, structures organization and determine the thyroid populations (see Fig. 2
and Fig. 3). T4 was assessed by IF and confocal images are included in both figures.
However, quantification of T4 levels in the medium was performed but not detected,
which might be due to the limit of detection of the T4 Elisa kit used in the study.

Did you perform T4 assay in culture medium?

R: As described above, despite the observation of T4 accumulation in the follicles, we
could not detect clear levels in the culture media, which can be due to the limit of
detection of the T4 Elisa kit used in the study.

Did you analyze proliferation from D47 to D58? And apoptosis?

R: Dear reviewer, please see Fig. 1e whereby BrdU uptake assay we describe the
proliferation rate from day 9 to day 58. Apoptosis was evaluated by IF using Caspase
3 Ab, and we did not observe apoptosis among thyroid cells (see above).

How long do T4-producing cells survive in culture in your model?

R: We have kept the cells in culture for 70 days and observe that the organoids keep
the expression of TG, TPO and accumulate TG-I in the lumen (see Supplementary
Fig. 3c).

How many stem cells are needed to reproduce a functional human thyroid
(11/12weeks)? Same if it is a proof of concept. Did you compare T4-immunostaining?
Quantification? (Fig 2d and compare with human thyroid tissue)

R: Dear reviewer, since it is not clear how to estimate the number follicles in the human
thyroid, we could use the transplantation experiment as example. Since for each
transplanted mice we used around 1,500 follicles and the generated tissue was
functional and almost recovered the levels in hypothyroid mice. We can estimate that
by using 6 wells of thyroid organoids we need initially 240,000 stem cells for EBs
generation. T4 immunostaining is compared at Fig. 4d, where grafted tissue is
compared to thyroid tissue.

Line 235: Did you perform NIS immunostaining on graft? And analyze of airway,
cardiovascular cells... by immunostaining on graft?

R: Dear reviewer, NIS staining was performed in organoids and grafted samples,
however the staining was not convincing since the localization was not corrected,
same staining was observed using human tissue, so we concluded that none of our
antibodies was efficiently staining NIS. Moreover, as shown at Fig. 4 f, ¹²³I uptake
images shows NIS presence since the cells could uptake iodine. The graft staining
with SMA demonstrates that all the non-thyroid cells are SMA+. We also did not
observe any E-Cadherin+ area without NKX2-1, TG staining, which can exclude the
presence of airway cells. In addition, Troponin T staining did not show any
cardiovascular cells.

Line 236: human tissue at each stage?

R: Dear reviewer, for the comparison we used adult thyroid tissue (see Methods: “RNA
extraction and quantitative real-time PCR” session).

Line 251: Did you perform TSH assay to validate the rescue?

R: Dear reviewer, as suggested, TSH was measured and the correlation with T4 levels
is demonstrated at Fig. 4i. In addition, we measured T3 levels and *Dio1* mRNA in the
liver, since *Dio1* is shown to be very sensitive to changes in TH levels (Fig. 4h and 4j;
Supplementary Fig. 6g).

Line 263: Could you compare your protocol and your results with those of Kurmanns?
(human cells)

R: Kurmann’s study shows a direct differentiation protocol to generate NKX2-
1+/PAX8+ thyroid follicular epithelial progenitors. This is the first aspect that differs
from our model, where we transiently induce NKX2-1 and PAX8 expression by
doxycycline incubation. In both cases, endoderm induction is promoted by Activin A
incubation, but Kurmann’s protocol includes an anteriorization step and 10 days of
incubation with FGF2 and BMP4 to induce thyroid specification. Also, the
differentiation/maturation period used is taken 22 days in their protocol, while in ours
it takes 49 days. Another important aspect is that our model uses FBS in the media
composition while the media used in Kurmann’s protocol is serum-free supplemented
with distinct compounds depending on the stage of the protocol. Since efficiency was
not evaluated in the study, we can compare to our organoid’s efficiency generation.
Finally, regarding the organization and functionality of the thyroid follicles we show the
3D follicular organization and accumulation of T4 inside the lumen, what is not
reported for Kurmann’s organoids.

Line 528: Could you precise “blocking solution”

R: Dear reviewer, since the assay used for proliferation assessment was changed in
the new version of paper, the sentence was removed, and the method rewritten
accordingly.

Line 549: Supp table 3? There is no Table 3.

R: Dear reviewer, sorry for the mistake, it referred to Supplementary table 3, which is
now corrected.

Line 676: What is MTG? explain please

R: Dear reviewer, MTG refers to Matrigel. It is now written when it appears for the first
time on the text (Page 4. L. 94).

Mann and Whitney test is more appropriate than t-test

R: Dear reviewer, all the analysis were performed again using non-parametric tests as
required.

References to be checked 7,8: incorrect references

R: Dear reviewer, the references are corrected accordingly.

15: Ogundipe et al.

R: Dear reviewer, thanks for noticing the mistake, the reference was corrected.

REVIEWER COMMENTS

Reviewer #1 (Remarks to the Author):

Romitti et al has provided several new data to address the original concern that I've raised and partially addressed the issues.

#1. Thanks for providing the immunostaining data to compare adult thyroid tissues. However, details are lacking to assess the data (for example what day?) It seems awkward that panel C claimed "follicular structure" is not entirely consistent with panel d T4 positive area which raises a concern that the follicular area is not entirely a thyroid tissue.

SI Fig6 H&E (and scRNAseq datasets) nuances this concern and it could be just the other epithelial cystic structure that often comes out from PSC-derivatives. It would be informative to show what's off-target tissue came out of these grafts and discuss differences/similarities from primary thyroid.

#2. Reproducibility concern is not addressed as the authors just added one different timepoint. But probably fine without these data, given a higher workload to get new sample and BMI done.

#3. Quantifying the human thyroid function after transplantation is not formally addressed. (I believe) ELISA-based assay didn't distinguish human versus mouse T4 . So in vivo quantification of T4 could be just a variation of RAI treatment or other indirect protection of the native thyroid.

To quantify organoid graft-derived tissues, it would be highly encouraged to look at graft-derived T4 production capability and discuss their function relative to original tissues.

Reviewer #2 (Remarks to the Author):

Romitti et al have made many corrections in this article with many experiments to improve its quality. The proofreading is complicated in the text and the figures as the article has been so much modified, but to be better. Mainly and among others, the essential dosage of T4 in the grafted animals.

The authors responded well to all questions and comments. They added more experiences than requested.

Some comments:

Fig 2D: the T4 immunostaining is unclear: could you improve this figure for the final picture, T4 is very weak? Or is there a problem for this picture?

T4 assay in medium: Have you tried concentrating the medium to assess T4?

Organoids at day 70 day: can they be grafted? Do they have larger follicles?

Nis antibody: have you tried Nancy Carrasco's NIS antibody?

Line 252: could you explain further "epithelial cells 1" vs 2: with KRT14 and KRT7 and AQP5

but why?

REVIEWER COMMENTS

Reviewer #1 (Remarks to the Author):

Romitti et al has provided several new data to address the original concern that I've raised and partially addressed the issues.

#1. Thanks for providing the immunostaining data to compare adult thyroid tissues. However, details are lacking to assess the data (for example what day?) It seems awkward that panel C claimed "follicular structure" is not entirely consistent with panel d T4 positive area which raises a concern that the follicular area is not entirely a thyroid tissue.

R: Figure 4 refers to D58 of organoids culture. Indeed, not all the thyroid follicles present in the grafted tissue produce detectable amount of T4, however, as showed at Fig 4C and D, epithelial cells presented in the grafted tissue express thyroid markers, such as NKX2-1, TG and TPO. Despite that a proportion of the follicles are not functional (T4+) after 5 weeks, it is very likely that the maturation and function is progressive and at this stage it might not have reached the full maturation.

SI Fig6 H&E (and scRNAseq datasets) nuances this concern, and it could be just the other epithelial cystic structure that often comes out from PSC-derivatives. It would be informative to show what's off-target tissue came out of these grafts and discuss differences/similarities from primary thyroid.

R: As described above, the epithelial structures present in the grafted tissue express markers such as NKX2-1, TG and TPO strongly indicating that those structures are thyroid tissue. Moreover, as described in the methods section, thyroid epithelial structures were enriched by filtration after dissociation, meaning that structures bigger than 100um were not used for the graft experiment. Also, the non-thyroid epithelial structures present within the organoids (lung airway mostly) are characteristically bigger than 100um, as you can appreciate at SI Fig 4 j and m. Moreover, as evidenced in our both single cell RNAseq, stem cells are not detected within the organoids at day 45 and 58, which reduces the chances of spontaneous generation of PSC-derived cystic structures after graft.

Concerning the off-target cells present within the grafted tissue, we detected by IF the presence of fibroblasts (SMA+) from which the origin is the human organoids (HNA+) and endothelial/vascular cells (CD31+) which form the blood vessels originated from the host (mouse cells). In addition, we have performed staining for lung markers (detected by scRNAseq in the organoids) and no cells were detected.

CD31 HNA NKX2-1 DAPI

SMA TG DAPI

#2. Reproducibility concern is not addressed as the authors just added one different timepoint. But probably fine without these data, given a higher workload to get new sample and BMI done.

R: Dear reviewer, as mentioned previously, due to the time and technical challenges to get additional “normal” thyroid tissue and more samples from our organoids, additional scRNAseq were not performed, except for day 58. However, we believe that the data present here, strongly supports the generation of human functional thyroid in vitro and in vivo. We also believe that understanding the mechanisms in each step of thyroid formation in vitro and the comparison to human thyroid tissue is valuable and can bring insights about mechanisms involved in thyroid development which are still unknown, and this is being addressed in ongoing projects in the lab.

#3. Quantifying the human thyroid function after transplantation is not formally addressed. (I believe) ELISA-based assay didn't distinguish human versus mouse T4 . So in vivo quantification of T4 could be just a variation of RAI treatment or other indirect protection of the native thyroid.

R. Dear reviewer, Indeed ELISA does not distinguish T4 from mouse or human, however, we can observe that in the RAI group, the animals do not show recovery or increase in T4 levels, what is clearly observed among the grafted animals (Fig 4G), similar results are obtained when T3 levels were analyzed. Moreover, as previously demonstrated at SI Fig 6F we can see progressive increase in T4 levels from week 4 to 5 after transplantation. Also, here we included additional data showing the T4 levels among the same animals, before transplantation, and after 4 and 5 weeks. This clearly shows that the transplant is the cause of increase in T4 levels, especially if we compare to RAI group at week 5 where the levels are similar to pre-transplantation (time point) levels of grafted group.

We also include here, the 123I uptake images from controls, RAI and grafted animals (respectively) at the end of the protocol, showing that no sign of thyroid tissue (123I uptake) is observed in the neck region of the RAI and grafted groups. Taken together these set of evidence strongly indicate that the thyroid native tissue is not recovered after RAI treatment and that the increase in THs is due to the organoids transplantation.

To quantify organoid graft-derived tissues, it would be highly encouraged to look at graft-derived T4 production capability and discuss their function relative to original tissues.

R. Dear reviewer, as suggested we have counted the T4-producing follicles (expressing at least one thyroid marker: NKX2-1, TG or TPO) inside the grafted tissue and we observed that around 58% of them have detectable T4. This information is now included in the text L.316-319. We also discuss that it suggests that there is an ongoing maturation/functional progress and that possibly soon it could reach full maturation. Also, we believe that comparing our transplanted organoids to adult thyroid tissue is not the most appropriated since we could consider our follicles as fetal thyroid, in which the accumulation and synthesis of THs is not comparable to adult tissue.

Reviewer #2 (Remarks to the Author):

Romitti et al have made many corrections in this article with many experiments to improve its quality. The proofreading is complicated in the text and the figures as the article has been so much modified, but to be better. Mainly and among others, the essential dosage of T4 in the grafted animals. The authors responded well to all questions and comments. They added more experiences than requested.

Some comments:

Fig 2D: the T4 immunostaining is unclear: could you improve this figure for the final picture, T4 is very weak? Or is there a problem for this picture?

R. Dear reviewer, due to the overlap with TG-I indeed the signal for T4 was not very visual, so we improved the quality of the image, now included in the Fig 3.

T4 assay in medium: Have you tried concentrating the medium to assess T4?

R. Dear reviewer, we have been working on T4 measurements by MS since ELISA kits are not sensitive enough for low amounts as we expect in our culture media. However, even trying to concentrate T4 we could not detect clear amount of T4 in the media. For the future the idea is to try to isolate T4 accumulate in the lumen of the follicles and also set up a protocol in order to collect without using the classical differentiation media (which might contain T4 in FBS) and/or PBS, since it is not compatible to the MS. Since it is quite challenging and time/culture consuming, we could not include clear measurements in this paper.

Organoids at day 70 day: can they be grafted? Do they have larger follicles?

R: The thyroid follicles kept in culture till day 70 show similar size and protein expression compared to day 58, which suggest that there is not major maturation/folliculogenesis occurring after day 58 in vitro. However, we have not grafted follicles from D70 into mice, so we cannot assert that it works as observed from D45. Also, using organoids from D45 is technically more advantageous since the culture period is considerably shorter and as we demonstrated the cells continue to mature in vivo.

Nis antibody: have you tried Nancy Carrasco's NIS antibody?

R: Dear reviewer, indeed we have tried some NIS antibodies provided by Nancy Carrasco but oppositely to mouse, in human organoids we don't see clear basal localization of the staining which also gave a lot of background making us to choose to do not show it (see pictures below). It might be that adjustments in the protocol are needed and we are working on that.

NIS (Carrasco) DAPI

Line 252: could you explain further “epithelial cells 1” vs 2: with KRT14 and KRT7 and AQP5 but why?

R: The clusters labeled "Epithelia cells" (1 and 2) express some lung cell type markers (such as KRT14, KRT7 and AQP5) but not a complete list of cell type markers otherwise expressed in the dataset. Thus, their differentially expressed genes could match many known cell types and they cannot be identified as belonging to any of them. The only label we can assign them with certainty is "epithelial" because they overexpress epithelial markers such as keratins (see the 50 first differentially expressed genes for clusters Epithelial cells 1 and Epithelial cells 2 below). The reasons to identify them as separate clusters (Epithelial cells 1 and Epithelial cells 2) and not giving them the same name (e.g. "Epithelial") are:

- 1) They are widely separated in the UMAP space
- 2) kNN clustering, even at low resolutions (resolution 0,1 in Seurat's FindClusters function) assigns them to 2 separate clusters
- 3) Their marker genes differ (see 50 first markers genes below where only 3 genes overlap)

Those 3 elements imply that they are two distinct cell populations that must be distinguished for downstream analyses such as ligand-receptor interactions.

Differentially expressed genes of cluster Epithelial cells 1 : MALAT1 S100A9 SPRR1B FABP5 MT-ND3 SPRR3 KRT5 MT-CO2 MT2A MT-ND4 MT-CO1 MT-CO3 MT-ATP6 NEAT1 MT-ND1 KRT13 MT-ND2 MT-CYB KRT15 KRT6A AGR2 COL3A1 S100P TMEM41B ORMDL1 TSPAN6 MTF2 TTC17 KRT17 MARCH6 SREK1 S100A6 ATP6V0D1 MIER1 BDH2 PLEKHJ1 MED10 PSMD10 POR RALY ATXN2 IARS2 USP48 VAMP2 WTAP PHF14 GSTK1 MKRN1 EMC4 GLOD4

Differentially expressed genes of cluster Epithelial cells 2 : DCD MUCL1 SCGB2A2 AZGP1 PIP SCGB1D2 SCGB1B2P AQP5 RPS4Y1 SFRP1 SAA1 LRRC26 ROPN1B STAC2 SAA2 TMEM213 CALML5 TFCP2L1 DNER KRT15 KRT7 SLC12A2 NCALD ADIRF RPL10 ZG16B CLDN10 RPL19 DBI PLA2R1 LGALS3 TSC22D1 COX4I1 RPS27A RBM3 SNORC FRZB PEBP1 RPL8 PPDPF GADD45B FAU SOD2 ITPR2 RPL30 NDUFB9 SLC26A2 KRT14 CD59 RPS5